



# Modelling symbiotic biological nitrogen fixation in grain legumes globally by LPJ-GUESS

Jianyong Ma[1], Stefan Olin[2], Peter Anthoni[1], Sam S. Rabin[1], Anita D. Bayer[1], Sylvia S. Nyawira[3], Almut Arneth[1,4]

[1]Institute of Meteorology and Climate Research-Atmospheric Environmental Research, Karlsruhe Institute of Technology, 82467 Garmisch-Partenkirchen, Germany

[2] Department of Physical Geography and Ecosystems Science, Lund University, 22362 Lund, Sweden

[3] International Center for Tropical Agriculture (CIAT), ICIPE Duduville Campus, P O Box 823-00621 Nairobi, Kenya

[4] Institute of Geography and Geoecology, Karlsruhe Institute of Technology, 76131 Karlsruhe, Germany

*Correspondence to*: Jianyong Ma (Jianyong.ma@kit.edu)

**Abstract.** Biological nitrogen fixation (BNF) from grain legumes is significant importance in global agricultural ecosystems. Crops with BNF capability are expected to support the need to increase food production while reducing nitrogen (N) fertilizer input for agriculture sustainability, but quantification of N fixing rates and BNF crop yields remains inadequate. Here we incorporate two legume crops (soybean and faba bean) with BNF into a dynamic vegetation model LPJ-GUESS (Lund-Potsdam-Jena General Ecosystem Simulator). The performance of this new implementation is evaluated against observations from a range of water and N management trials. LPJ-GUESS generally captures the observed response to these management practices on legume biomass production, soil N uptake and N fixation, despite some deviations from observations in some cases. Globally, the simulated BNF is dominated by soil moisture and temperature, as well as amounts of N fertilizer addition. Annual inputs through BNF are modelled to be 11.6±2.2 Tg N for soybean and 5.6±1.0 Tg N for all pulses, with a total fixation of 17.2±2.9 Tg N yr$^{-1}$ for all grain legumes during the period 1981-2016 on global scale. Our estimates show a good agreement with some previous statistical estimates but are relatively high compared to some estimates for pulses. This study highlights the importance of accounting for legume N fixation process when modelling C-N interactions in agricultural ecosystems, particularly when it comes to account for the combined effects of climate and land-use change on global terrestrial N cycle.

## 1 Introduction

The agricultural sector is the main contributor to anthropogenic nitrous oxide ($N_2O$) emissions (Reay et al., 2012; Tian et al., 2020) as well as a key nitrate pollution source to freshwater systems (Moss, 2008), mostly due to the intensive use of synthetic nitrogen (N) fertilizer and animal manure (Lu and Tian, 2017). This trend has been amplified by the expansion of agricultural land to provide food for a growing population and changing dietary patterns (FAO, 2018). The use of crops with biological N fixation (BNF) capability in agriculture has been discussed as one option to address the conflict between the need to increase food production and the associated environmental problems of N loss (Becker, et al., 1995; Fageria, 2007; Northup and Rao, 2016). N-fixing crops, like grain and forage legumes, not only provide protein-rich food for the human population and farmed animals (Voisin et al., 2014; Stagnari et al., 2017), but they are also directly useable as "green manure" reducing the amount of chemical N fertilizer required in agricultural systems (Liu et al., 2011; Meena et al., 2018).

Soybean (*Glycine max* L.), with its countless and varied uses, is now one of the most widely grown crops in the world because of attractive cash return from its grain yield (FAOSTAT, 2021). Concerns about the sustainability of soybean production exist in particular because of its links to deforestation and loss of native vegetation in the Amazon and other areas of South America





(Fehlenberg et al., 2017; Heilmayr et al., 2020). Other grain legumes, such as faba bean (*Vicia faba* L.), chickpea (*Cicer arietinum* L.) and cowpea (*Vigna unguiculata* L.), play an important role in improving soil quality as "green manure" when they are rotated or used as intercrops between cereals depending on the region (Williams et al., 2014; Denton et al., 2017). In comparison to non-legume

plants, using legumes as "green manure" is more effective to build up or maintain soil fertility, as they not only increase soil organic matter when adding their biomass to soils, but also add extra N into the soil resulting from their symbiotic association with bacteria (Peoples et al., 2009; Ciampitti and Salvagiotti, 2018). The enriched soil N and soil organic carbon contents jointly support growth and productivity in subsequent crops (Jensen et al., 2012; Hajduk et al., 2015). Much experimental evidence has indicated that grain legume biomass increases linearly with increasing BNF rate (Salvagiotti et al., 2008; Unkovich et al., 2010; Córdova et al., 2019) and

that the N benefit to soil fertility from green manure is closely correlated to the N fixation capacity assuming that the entire legume plant is tilled into the soil (Fageria, 2007; Meena et al., 2018). Estimating the rate of BNF is thus important not only for an accurate prediction of grain legume production but also for a better understanding of where and to what degree N loss (i.e., N leaching and gaseous N emission) in cropland systems can be reduced by partially or fully replacing chemical N fertilizer with legume green manure.

Although grain legumes' BNF rates can be measured at field sites and in controlled environments, ecological models are needed for understanding and quantifying the rate of BNF on larger spatial scales and longer temporal perspectives. In many process-based crop models, a common method of representing BNF is to use a pre-defined potential or maximum N fixation rate that is adjusted by limiting environmental factors (Liu et al., 2011). The potential N fixation rate is then estimated either from plant nodule, root, and aboveground biomass (e.g., CROPGRO, Boote et al., 2008; STICS, Corre-Hellou et al., 2009; SPACSYS, Wu et al., 2020) or from

plant N demand status (e.g., EPIC, Cabelguenne et al., 1999; APSIM, Robertson et al., 2002), varying with plant life cycle. Environmental constraining factors, such as soil temperature, water availability, soil mineral N concentration, and plant growth stage are mostly taken into account (Liu et al., 2011; Chen et al., 2016). The big challenge in modelling legume BNF is that the process of symbiotic N fixation is always accompanied by the cost of fixed total photosynthetic carbon (C) to maintain legume symbioses growth, activity, and reserves, which may be around of 4-16% of C (Kaschuk et al., 2009). Such a photosynthetic consumption strength would

result in productivity loss if photosynthesis rate does not increase to compensate for the cost (Kaschuk et al., 2010). In most models C cost mechanisms have not been implemented into N fixation consistent with the assumption that the plant N uptake from soils does not cost carbon (e.g., EPIC, Cabelguenne et al., 1999; APSIM, Robertson et al., 2002; STICS, Corre-Hellou et al., 2009; LPJmL, Von Bloh et al., 2018; SPACSYS, Wu et al., 2020), despite many field experiments demonstrating that energy consumption required for BNF is far larger than soil mineral N uptake (Ryle et al., 1979; Harris et al., 1985; Macduff et al., 1996). In several other models, root

substrate C concentration was adopted as an alternative to represent the C demand of N fixation (e.g., Hurley Pasture Model, Thornley and Cannell, 2000; TEM, Yu and Zhuang, 2020). Only a few models assume that such a consumption can be assessed directly against C acquired in photosynthesis, in which the C cost per unit of fixed N is defined as either a constant of 6 kg C kg N$^{-1}$ (CROPGRO, Boote et al., 2008; O-CN, Meyerholt et al., 2016) or a dynamic function of soil temperature ranging between 7.5 and 12.5 kg C kg N$^{-1}$ (FUN, Fisher et al., 2010; Houlton et al., 2008).

The global production and consumption of grain legumes have greatly increased over recent decades (FAOSTAT, 2021). Accurately representing and quantifying the dynamic process of biological N fixation in models is important for better understanding grain legumes' contribution to food security and agriculture sustainability, particularly in the context of global environmental change. However, because of inadequate information on the environment and crop management, as well as the missing or incomplete BNF mechanism in models (e.g., C cost as mentioned above), current simulation of grain legume N fixation and its yield is still very weak,

especially when it comes to global scale modelling.





Thus, in this study, accounting for the importance of soybean in overall agriculture and trade, and the higher N fixation capacity of faba bean compared to other pulses (Peoples et al., 2009; Unkovich et al., 2010; Denton et al., 2017; Liu et al., 2019), we implement these two grain legumes with BNF into a process-based vegetation model (LPJ-GUESS, Smith et al., 2014; Olin et al., 2015). Processes are added to LPJ-GUESS to estimate the symbiotic relationship between legumes and bacteria, also taking into account the plant C cost of BNF. Model results are extensively evaluated with worldwide site-level observed data and compared against country-level yield statistics, as well as continent-level BNF rates. The model-based and large-scale quantification of the N fixation capacity in legumes provides a scientific foundation for predicting present and future N cycle in agro-ecosystems, allowing recommendations for fertilizer N application under different climatic conditions in legume-based farming systems.

## 2 Methods

### 2.1 Model description

LPJ-GUESS is a process-based dynamic vegetation model that simulates carbon and nitrogen (C-N) dynamics at scales ranging typically from regionally to global (Smith et al., 2014). The model represents vegetation and soil dynamic processes and their interactions in response to changes in the environment and management, such as climate, $CO_2$ concentration, soil physical properties, N deposition and N fertilization. Three land-use types are included in the model: natural vegetation, pasture and cropland. Vegetation on natural land are represented as the establishment, growth, and mortality of 12 plant function types (PFTs). Pastures are simulated by competing C3 and C4 grasses, in which 50% of above-ground biomass are annually harvested to account for the effects of grazing (Lindeskog et al., 2013). Crops in LPJ-GUESS are described by crop functional types (CFTs), which differ in their C assimilation allocation scheme, morphological traits, and heat sum requirement for growth. At present, four CFTs are represented in the C-N version of LPJ-GUESS: two temperate C3 crops with sowing carried out in spring and autumn, a tropical C3 crop (representing rice), and a C4 crop (representing maize). The recent representation of crops includes the incorporation of soil N transformation (Olin et al., in prep; see also Tian et al., 2020) together with a C-N allocation for crops operating on a daily time step (Lindeskog et al., 2013; Olin et al., 2015). Cropland management options for global-scale application include variable sowing and harvest dates, irrigation, tillage, N application, cover crop grass between the main growing seasons, and residue retention (Pugh et al., 2015; Olin et al., 2015). In this study, soybean is simulated as one additional crop because of its large importance as a food, fodder and oil crop, and faba bean as a second as a representation of pulses more generally. The model schematic and other calculations including the C cycle and the N cycle follow an earlier version of LPJ-GUESS (Smith et al., 2014; Warlind et al., 2014; Olin et al., 2015).

### 2.2 Updated daily carbon allocation parameters

Similar to most ecosystem and crop models, LPJ-GUESS adopts crop-specific accumulated heat requirements to model plant growth development (Lindeskog et al., 2013). To better represent C and N allocation in various phenological phases, Olin et al. (2015) defined crop development stage by considering the effects of temperature, vernalisation days and photo-period, following Wang and Engel (1998). In this study, we assumed that grain legume development stage is linearly correlated to its accumulated heat units according to the field-based soybean experiments described in Irmak et al. (2013). It is estimated as:

$$DS = \begin{cases} a_{veg} + b_{veg} \times fphu & (fphu \leq \text{fphu}_{anthesis}) \\ a_{rep} + b_{rep} \times fphu & (fphu > \text{fphu}_{anthesis}) \end{cases} \tag{1}$$

where $DS$ is crop development stage, ranging from 0 to 2 ($DS$=0, sowing; $DS$=1, flowering; $DS$=2, harvest); $fphu$ is the fraction of today's accumulated heat units to total heat requirement; $\text{fphu}_{anthesis}$ is the threshold of $fphu$ when anthesis starts, below (above) which crop growth belongs to the vegetative (reproductive) stage; and a and b are the linear regression coefficients, varying between the

vegetative and reproductive phases. The values of a and b, and the crop-specific base temperature (°C) to estimate the accumulated heat units are both given in Table S1 (in the Supplement).

Daily fraction of assimilate allocation to leaves, stems and roots is an important process before storage organs are formed. The assimilate invested in roots can help crops overcome water or nutrient limitation when they suffer from stress in the vegetative stage,
whereas new assimilate invested in leaves generally gives a highly efficient return from the photosynthesis product (Penning de Vries et al., 1989). Unlike cereal crops, nodulated plants, particularly soybeans, are more likely to achieve a higher photosynthesis rate and delay leaf senescence due to the continued N supply from biological N fixation (Abu-shakra et al., 1978; Kaschuk et al., 2010). A precise representation of assimilate partitioning to the plant organs when modelling BNF in grain legumes is especially important considering the high C cost from fixing N from the atmosphere. Productivity loss would be simulated if leaf photosynthesis rate would
not increase to compensate for the costs (Macduff et al., 1996; Kaschuk et al., 2009).

Following Olin et al. (2015), relationships between assimilate allocation to legume organs were established based on the data from Penning de Vries et al. (1989) and Boote et al. (2002). We fitted the allocation functions using Richards logistic growth curve (Eq. (2), Richards, 1959) to model the allocation to each organ dynamically and separately. For each allocation function $f_i$ (see Eqs. (3)-(5) below),

$$f_i = a_i + \frac{b_i - a_i}{1 + e^{-C_i \times (DS - d_i)}}$$  (2)

where $DS$ is crop development stage and $a_i$, $b_i$, $c_i$, $d_i$ are fitting coefficients for the three functions (specific values given in Table S1). Maintaining BNF in the reproductive stage (i.e., after anthesis; $DS > 1$) would reduce the flow of carbon assimilation to storage organs. We adjusted the allocation functions from Olin et al. (2015) so that the model allowed a dynamic adaptation of the allocation to grain over the seed-filling period in response to BNF cost (see Eqs. 3-5 for details).

**(1) Yield vs. the whole plant**

After anthesis ($DS > 1$), most assimilates are allocated and retranslocated from the vegetative organs to the grains. During the late seed-filling period ($DS \geq d_1$, see Eq. (3)), we assumed that the fraction of carbon allocated to yield would increase to partly compensate the productivity loss caused by spending on N fixation, with the cost of reducing the flow of carbon to leaves and stem (see Eq. (4)). We established the ratio of the allocation to yield relative to the whole plant as:

$$f_1 = \frac{P_{yield}}{P_{veg} + P_{yield}} = \begin{cases} a_1 + \dfrac{b_1 - a_1}{1 + e^{-c_1 \times (DS - d_1)}} & DS < d_1 \\ \left( a_1 + \dfrac{b_1 - a_1}{1 + e^{-c_1 \times (DS - d_1)}} \right) \times (1 + P_{BNFcost}) & DS \geq d_1 \end{cases}$$  (3)

where $P_{yield}$ and $P_{veg}$ are the fraction of carbon allocated to yield and vegetative organs, respectively, ranging from 0 to 1; $P_{BNFcost}$ is the
proportion of NPP used for BNF to today's total NPP; $d_1$ is the fitting coefficient, representing the $DS$ of maximum growth rate of grain (d=1.41 for soybean and 1.46 for faba bean, see Table S1).

**(2) Leaf vs. shoot vegetative organs**

Similarly, the ratio of leaf vs. shoot vegetative allocation is specified as:

$$f_2 = \frac{P_{leaf}}{P_{veg} - P_{root}} = \begin{cases} a_2 + \dfrac{b_2 - a_2}{1 + e^{-c_2 \times (DS - d_2)}} & DS < d_1 \\ \left( a_2 + \dfrac{b_2 - a_2}{1 + e^{-c_2 \times (DS - d_2)}} \right) - P_{BNFcost} & DS \geq d_1 \end{cases}$$  (4)



where $P_{leaf}$ and $P_{root}$ are the fraction of carbon allocated to leaf and root, respectively. The fitting function of leaf vs. shoot vegetative

organs in soybean is given in Fig. 1a.

### (3) Root vs. vegetative organs

When a plant experiences water or nutrient stress, it invests more assimilate to roots relative to shoot vegetative organs (Penning de Vries et al., 1989). We implemented dynamic increases in the allocation to roots during the late seed-filling period to help legumes cope with the C loss from BNF cost, and established the relationship between the allocation to root and that to vegetative organs as:

$$f_3 = \frac{P_{root}}{P_{veg}} = \begin{cases} a_3 + \dfrac{b_3 - a_3}{1 + e^{-c_3 \times (DS - d_3)}} & DS < d_1 \\ \left( a_3 + \dfrac{b_3 - a_3}{1 + e^{-c_3 \times (DS - d_3)}} \right) + (1 - f_1) \times P_{BNFcost} & DS \geq d_1 \end{cases} \quad (5)$$

In addition, carbon partitioning to vegetative organs ($P_{veg}$) can be calculated by subtracting the reproductive allocation (i.e., $P_{yield}$) from the whole plant as:

$$P_{veg} + P_{yield} = 1 \Rightarrow P_{veg} = 1 - P_{yield} = 1 - f_1 \quad (6)$$

Finally, we can achieve dynamic carbon allocation to the plant organs over the growing season by combining Eqs. (3)-(6):

$$\begin{cases} P_{yield} = f_1 \\ P_{leaf} = f_2 \times (1 - f_1) \times (1 - f_3) \\ P_{stem} = P_{veg} - P_{root} - P_{leaf} = (1 - f_1) \times (1 - f_2) \times (1 - f_3) \\ P_{root} = f_3 \times (1 - f_1) \end{cases} \quad (7)$$

Partitioning functions are plotted for soybean in Fig. 1b and for faba bean in Fig. S1 (in the Supplement). Significant difference in

allocation patterns can exist between cultivars. Compared to cereals (Olin et al., 2015), we found that grain legumes are more likely to allocate more assimilate to leaves not only in partitioning proportion but also in the length of allocation time, probably corresponding to their higher leaf activities in response to N fixation (Kaschuk et al., 2010).

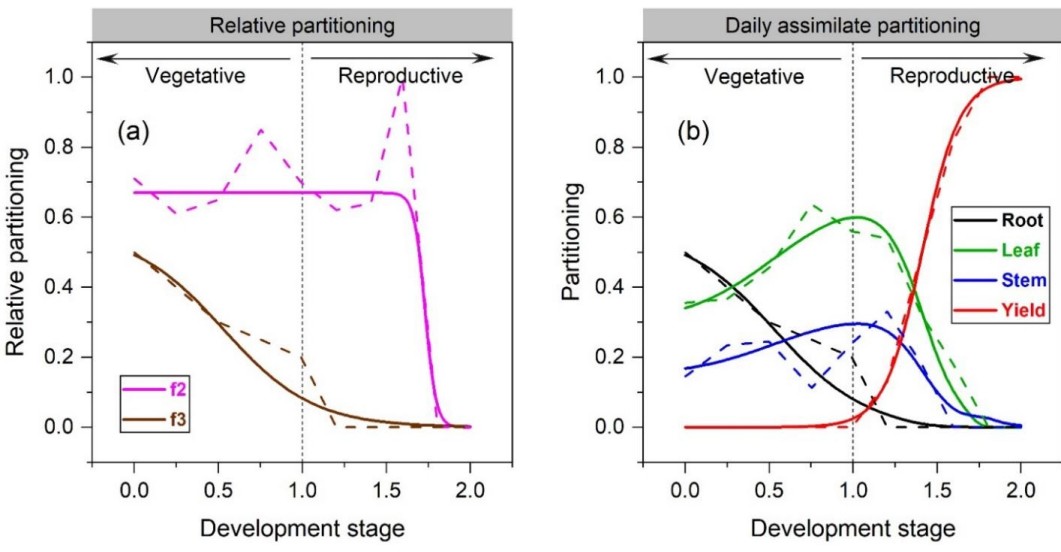

**Figure 1.** The organ's relative allocation (a) and assimilate partitioning (b) to roots, leaves, stem and yields for soybean. Solid lines represent the

fitted Richards functions in this study and dashed lines are the allocation scheme from Penning de Vries et al. (1989). $f_2$ in (a) denotes leaf relative allocation to shoot vegetative organs (Eq. (4)), whereas $f_3$ is root relative allocation to vegetative organs (Eq. (5)).





### 2.3 Representation of BNF

Fixing N from the atmosphere and N uptake from soils are two N sources for grain legumes to meet their total plant N demand. The latter has a higher priority for plants because the process is less energy-consuming than N fixation (Ryle et al., 1979; Macduff et al., 1996). Following on this idea, in LPJ-GUESS, N fixation will only be triggered when the following two assumptions are valid at the same time (Fig. 2): (1) if today's plant growth still suffers from N-limitation after N uptake from soils (i.e., the N deficit, plant N demand minus soil N uptake, is greater than zero). The plant will then be allowed to fix N from the atmosphere to fill the N deficit. (2) Since N fixation is strongly related to photosynthetic assimilate due to its high energy consumption, BNF in the model is assumed to take place only when today's NPP is positive, so that adequate C supply can be provided to meet the BNF cost.

Modelling the BNF rate is adapted from previously published methods (CROPGRO, EPIC, APSIM etc., see Liu et al., 2011), in that it considers (1) the potential N fixation rate, (2) the limitation of temperature, (3) soil water status, and (4) the crop growth stage as:

$$N_{fix} = N_{fixpot} \times f_T \times f_W \times f_{DS} \tag{8}$$

where $N_{fix}$ is the N fixation rate; $N_{fixpot}$ is the potential N fixation rate; and $f_T$, $f_W$, $f_{DS}$ are limitations (ranging 0 to 1) on BNF by soil temperature, soil water availability, and crop development stage function, respectively.

The definition of potential N fixation rate in some studies is based on the strong relationship between N fixation and either nodule size/biomass (Weisz et al., 1985; Voisin et al., 2003) or root dry matter (Soussana et al., 2002; Voisin et al., 2007). Due to the difficulties in measuring both nodules and roots in the field directly, some studies also adopt shoot biomass to replace nodule/root biomass, based on the empirical relationship between these two variables (Yu et al., 2002; Corre-Hellou et al., 2009; Wu et al., 2020). In our implementation, since the nodulation process of legumes has not yet been implemented in LPJ-GUESS, $N_{fixpot}$ is assumed to be proportional to root dry matter:

$$N_{fixpot} = \mathrm{N_{maxfixpot}} \times DM_{root} \tag{9}$$

where $\mathrm{N_{maxfixpot}}$ is the maximum nitrogen fixation rate of roots (g N g$^{-1}$ root DM) and $DM_{root}$ is root dry matter (g root DM m$^{-2}$). Since the experimental parameter $\mathrm{N_{maxfixpot}}$ is strongly related to the effectiveness of rhizobial strains, and varies widely between species and sites, it is not easy to obtain the parameter for each legume crop. In this study, we assume that legumes are either inoculated or there are high enough populations of strains in the soil so that $\mathrm{N_{maxfixpot}}$ is not constrained by the effectiveness of rhizobia. Here $\mathrm{N_{maxfixpot}}$ is assumed to be a constant as 0.03 g N g$^{-1}$ root DM for both grain legumes as a moderate value taken from the literature (Soussana et al., 2002; Eckersten et al., 2006; Boote et al., 2008).

Soil temperature is a controlling factor for both microbial activities and plant growth. For soybean, 20-35°C has been found to be optimal for nitrogenase activity and for faba bean the optimal soil temperature can range between 16-25°C (Boote et al., 2008). The influence of soil temperature on legume BNF is represented in the model as a four-threshold-temperature function:

$$f_T = \begin{cases} 0 & (T < \mathrm{T_{min}} \ or \ T > \mathrm{T_{max}}) \\ \frac{T-T_{min}}{T_{optL}-T_{min}} & (\mathrm{T_{min}} \leq T < \mathrm{T_{optL}}) \\ 1 & (\mathrm{T_{optL}} \leq T \leq \mathrm{T_{optH}}) \\ \frac{T_{max}-T}{T_{max}-T_{optH}} & (\mathrm{T_{optH}} < T \leq \mathrm{T_{max}}) \end{cases} \tag{10}$$

where $T$ is soil temperature (°C) at 25 cm depth, $\mathrm{T_{min}}$ ($\mathrm{T_{max}}$) is the minimum (maximum) temperature below (above) which N fixation stops, and $\mathrm{T_{optL}}$ and $\mathrm{T_{optH}}$ are the lower and higher optimal temperatures within which N fixation is not limited by temperature. The values of these four temperature thresholds vary among legume crops and are given in Table 1.

In addition to temperature, the soil water content is a major factor controlling the rate of N fixation (Srivastava and Ambasht, 1994). Too little or too much water dramatically inhibits BNF due to impacts of drought stress and oxygen deficit, respectively, on nodule



nitrogenase activity (Marino et al., 2007). A linear water-limitation function is incorporated into LPJ-GUESS (Wu and McGechan, 1999), and is represented as:

$$f_W = \begin{cases} 0 & (W_f \leq W_a) \\ \varphi_1 + \varphi_2 \times W_f & (W_a < W_f < W_b) \\ 1 & (W_f \geq W_b) \end{cases} \tag{11}$$

where $W_f$ is relative soil water content in the top soil layer (0-50cm), ranging from 0 to 1; $\varphi_1$ and $\varphi_2$ are empirical coefficients; $W_a$ is the threshold of $W_f$ below which N fixation is fully restricted by soil water deficit and $W_b$ is the value above which N fixation is not inhibited by soil water content. The values of the parameters are shown in Table 1.

The influence of plant growth stage on legume BNF rate is taken into account in very few models; the process is generally stopped forcibly after the crop reaches a certain development stage. For example, in the CROPGRO model (Boote et al., 2008), N fixation in soybean starts in the early vegetative stage and continues until the end of physiological maturity, whereas it ceases at the middle of the seed-filling period in the EPIC model (Cabelguenne et al., 1999). In this study, a more specific function, similar to the temperature response function, is implemented to the BNF scheme to represent the variation of N fixation with the course of legume life cycle:

$$f_{DS} = \begin{cases} 0 & (NDS < \text{NDS}_{min} \text{ or } NDS > \text{NDS}_{max}) \\ \frac{NDS - \text{NDS}_{min}}{\text{NDS}_{optL} - \text{NDS}_{min}} & (\text{NDS}_{min} \leq NDS < \text{NDS}_{optL}) \\ 1 & (\text{NDS}_{optL} \leq NDS \leq \text{NDS}_{optH}) \\ \frac{\text{NDS}_{max} - NDS}{\text{NDS}_{max} - \text{NDS}_{optH}} & (\text{NDS}_{optH} < NDS \leq \text{NDS}_{max}) \end{cases} \tag{12}$$

where $NDS$ is normalized crop development stage, ranging from 0 to 1 (0, sowing; 0.5, flowering; 1, harvest); $\text{NDS}_{min}$ is the time before which there is no N fixation due to inadequate nodulation; $\text{NDS}_{max}$ is the time after which N fixation suspends due to nodule senescence; and $\text{NDS}_{optL}$ and $\text{NDS}_{optH}$ define the period within which legume BNF rate is not inhibited by development stage. The values of the parameters for two grain legumes are derived from the literature and listed in Table 1.

In addition to the environmental limitation factors, the amount of daily NPP also affects N fixation in the model. The NPP requirement for BNF cost is computed based on the estimated N fixation rate ($N_{fix}$, Eq. (8)) by multiplying the C cost per unit fixed N, which is assumed to be a fixed value of 6 g C g$^{-1}$ N fixed as a moderate value taken from previous studies (Ryle et al., 1979; Patterson and Larue, 1983; Boote et al., 2008; Kaschuk et al., 2009). The NPP cost to maintain BNF is released as $CO_2$ to the atmosphere and modelled as part of the autotrophic respiration of the soil (Fig. 2). Since the fixed N is partly transported to plant leaves and continues

to support the photosynthesis activity, the plant may further get C profits from the investment of BNF by enhancing the leaf N content that optimizes the carboxylation capacity ($V_{max}$) (Kull, 2002). Following on this idea, another assumption adopted in this study is that at maximum 50% of today's NPP can be used for N fixation before the crops reach the development stage of grain maximum growth rate ($DS < d_1$, see Eq. (13)). After this the maximum proportion of today's NPP used for BNF cost is dynamically reduced and assumed to be the fraction of carbon allocation to leaves and stem:

$$MAX_{BNFcost} = \begin{cases} 0.5 & DS < d_1 \\ P_{leaf} + P_{stem} = (1 - f_1) \times (1 - f_3) & DS \geq d_1 \end{cases} \tag{13}$$

where $MAX_{BNFcost}$ is the maximum proportion of today's NPP used for N fixation, varying from 0-0.5; $P_{leaf}$ and $P_{stem}$ are the fraction of carbon (i.e., NPP) allocated to leaf and stem, respectively (see Eq. (7) for details). A flowchart of the BNF scheme in LPJ-GUESS is shown in Fig. 2.

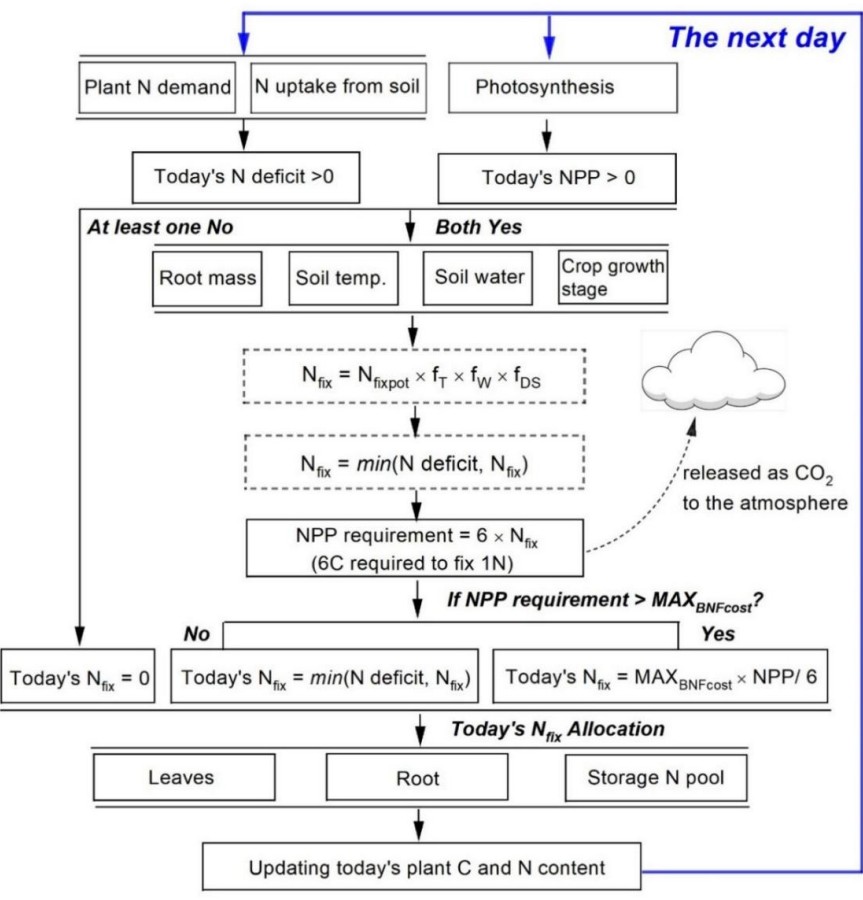

**Figure 2.** Representation of the N fixation route used in grain legumes in LPJ-GUESS. Today's N deficit is calculated as the difference between
plant N demand and soil mineral N uptake. $N_{fix}$ in dotted boxes are intermediate values.





**Table 1.** Overview of BNF related variables and parameters used in the model for soybean and faba bean.

| Parameter | Description | Soybean | Faba bean | Unit | Reference |
|---|---|---|---|---|---|
| N deficit | plant N demand minus soil N uptake | dynamic | dynamic | g N m$^{-2}$ d$^{-1}$ | |
| NPP | net primary productivity | dynamic | dynamic | g C m$^{-2}$ d$^{-1}$ | |
| $N_{maxfixpot}$ | maximum nitrogen fixation rate of roots | 0.03 | 0.03 | g N g$^{-1}$ root DM | Soussana et al., 2002; Eckersten et al., 2006; Boote et al., 2008 |
| $DM_{root}$ | root dry matter | dynamic | dynamic | g root DM m$^{-2}$ | |
| C cost | Carbon cost per unit fixed N | 6 | 6 | g C g$^{-1}$ N fixed | Ryle et al., 1979; Boote et al., 2008; Kaschuk et al., 2009 |
| T | soil temperature at 25cm depth | dynamic | dynamic | °C | |
| $T_{min}$ | the minimum temperature for the start of N fixation | 5 | 1 | °C | Boote et al., 2008 |
| $T_{optL}$ | lower bound of optimal temperature for N fixation | 20 | 16 | °C | Boote et al., 2008 |
| $T_{optH}$ | upper bound of optimal temperature for N fixation | 35 | 25 | °C | Boote et al., 2008 |
| $T_{max}$ | the maximum temperature for the stop of N fixation | 44 | 40 | °C | Boote et al., 2008 |
| $W_f$ | relative soil water content in the top layer (0-50cm) | dynamic | dynamic | - | |
| $W_a$ | lower bound of water content below which N fixation is fully limited by soil water deficit | 0.2 | 0 | - | Robertson et al., 2002 |
| $W_b$ | upper bound of water content above which N fixation is not inhibited by water content | 0.8 | 0.5 | - | Robertson et al., 2002 |
| $\varphi_1$ | coefficient of soil water content | -0.33 | 0 | - | Robertson et al., 2002 |
| $\varphi_2$ | coefficient of soil water content | 1.67 | 2 | - | Robertson et al., 2002 |
| NDS | normalized crop development stage | dynamic | dynamic | - | Wang and Engel, 1998 |
| $NDS_{min}$ | the minimum development stage for the start of N fixation | 0.1 | 0.1 | - | Bouniols et al., 1991 |
| $NDS_{optL}$ | lower bound of development stage for N fixation | 0.3 | 0.3 | - | Bouniols et al., 1991 |
| $NDS_{optH}$ | upper bound of development stage for N fixation | 0.7 | 0.6 | - | Bouniols et al., 1991 |
| $NDS_{max}$ | the maximum development stage for the stop of N fixation | 0.9 | 0.8 | - | Bouniols et al., 1991 |



### 2.4 Experimental set-up

Field-based data from the literature, together with global yield statistics from legume-producing countries and region-level N fixation data from published sources, were compared to model runs to examine performance in simulating yields and BNF rate from site scale to a larger region.

In order to build up cropland soil C and N pools, all simulations were initialized with a 500-year spin-up using atmospheric $CO_2$ from 1901 combined with repeating detrended 1901-1930 climate from GSWP3-W5E5 (Dirmeyer et al., 2006; Lange, 2019; Cucchi et al.,

2020). The cropland fraction linearly increased from zero to the first historic value (1901) during the last 30 years of spin-up. Monthly atmospheric N deposition ($NH_x$, $NO_y$) was used as simulated by CCMI (NCAR Chemistry-Climate Model Initiative). The value was interpolated to $0.5° \times 0.5°$ from the original resolution ($1.9° \times 2.5°$) to match the resolution of the climate data (Tian et al., 2018). Below, the set-up of the different experiments is explained in detail.

### 2.4.1 Model evaluation at site scale

To evaluate the model's ability to simulate BNF rate and yields, field-based N fixation trials with detailed measurements of soil N uptake, biomass and N mass allocation were collected from the published literature. This dataset comprised 17 soybean and 7 faba bean sites located between ~33°S and ~53°N (Fig. 3). In these trials, BNF response to various management practices (such as N fertilizer addition and irrigation) were investigated. Details about these sites—their geographic coordinates, BNF trials and the years of available data, as well as corresponding site-specific plant traits (e.g., specific leaf area, and grain C:N ratio)—are provided in

Table S2.

In some field experiments, BNF rate and/or soil N uptake are not directly reported in the literature, we estimated these values as:

$$\begin{cases} BNF_{obs} = \%Ndfa \times N_{plant} \\ SoilNuptake_{obs} = (100 - \%Ndfa) \times N_{plant} \end{cases} \qquad (14)$$

where *%Ndfa* is the proportion of plant N derived from the atmosphere (ranging 0-100), representing the contribution of N fixation to the plant total N uptake; $N_{plant}$ is the amount of N accumulated in the plant (kg N ha$^{-1}$), and defined as either the shoot or the whole plant N mass, depending on the measurement method adopted in the experiment.

In general, grain yields, plant tissues' dry mass and N mass, together with %Ndfa, soil N uptake and N fixation are widely-measured variables in the field-based BNF trials (see Table S2). These data were chosen as our target variables used for model evaluation. In addition, to convert plant C mass to dry matter, a conversion factor of 2.0 was used (Smith et al., 2014). Dry weight was converted to wet weight by assuming a water fraction of 0.13 in the grain legumes (Córdova et al., 2019).

Since specific leaf area (SLA) and target grain C:N ratio play a very important role in determining N uptake and N retranslocation to

grain during seed-filling in the model (Olin et al., 2015), we implemented two simulations to explicitly explore model performance across all sites. For 'site-specific' simulations, the reported SLA and grain C:N ratio listed in Table S2 were adopted for the simulation (for sites for which these were available). For 'global-uniform' parameter simulations, SLA was set to 40 and 45 m$^2$ kg$^{-1}$ C (Penning de Vries et al., 1989), and target grain C:N ratio was represented as a constant of 8 for soybean and 10 for faba bean, respectively (Kattge et al., 2020). These values were also used for global-scale simulations.

Gridded daily climate data of air temperatures (maximum, minimum and mean), precipitation, and solar radiation were used from GSWP3-W5E5 (Dirmeyer et al., 2006; Lange, 2019; Cucchi et al., 2020), chosen for the $0.5° \times 0.5°$ grid cell representative for each experimental site. There was no information on land use and management practices in years preceding the experiments at most sites. Therefore, to maintain soil N and C pools in equilibrium after model spin-up, we decided to implement a common cropping system of maize-legume rotation annually from 1901 to the year before the trials start, with no N fertilizer applied to legumes. Over the trials

period, the management practices were implemented according to information provided in the literature (Table S2). In addition, site-



specific soil physical properties, such as fractions of sand, silt and clay, were also used as forcing to further compute corresponding soil water characteristics in the model (Olin et al., 2015).

### 2.4.2 Global yields and BNF rate

To evaluate the model's ability to simulate legume yields and BNF on a larger scale, national crop yield statistics from FAOSTAT (http://www.fao.org/faostat/en/#data/QC, last access: 9 May 2021) were collected and compared with modelled output. Furthermore, Peoples et al. (2009) divided N fixation data for widely-grown legume crops collated from a range of published sources into different geographical regions. In order to compare our simulated BNF with the literature-based records, each simulated $0.5°\times0.5°$ grid cell was classified to be in one of the ten regions given in Table 1 in Peoples et al. (2009) (Fig. S2).

For regional comparison, the modelled gridded yield and BNF rate were aggregated to national and continental scales, respectively,
using information of crop-specific cover area on spatial pattern:

$$Var_{region} = \frac{\sum_{i=1}^{n} [(Var_{rain})_i \times (Area_{rain})_i + (Var_{irri})_i \times (Area_{irri})_i]}{\sum_{i=1}^{n} [(Area_{rain})_i + (Area_{irri})_i]} \tag{15}$$

where $Var$ is yield or BNF rate; $Var_{region}$ is the aggregated result in a given region; $i$ is the gridcell number in that region, ranging from 1 to n; $Var_{rain}$ and $Var_{irri}$ represent the modelled yield or BNF rate under rain-fed and irrigated conditions, respectively; $Area_{rain}$ and $Area_{irri}$ are the crop-specific rain-fed and irrigated areas used in simulations, respectively.

As land use/land cover input, data from LUH2 (Land-Use Harmonization 2, Hurtt et al., 2020) with fractions of cropland, pasture,
natural vegetation at each grid cell was adopted, spanning from 1901 to 2014 in 0.5° resolution. The fractional cover of different crop species was derived from MIRCA (Monthly Irrigated and Rain-fed Crop Areas, Portmann et al., 2010). Since no detailed information was available on the growth-distribution of faba bean, the 'pulse' fraction in MIRCA was used as input instead, and 'pulses' country-level yield statistics provided by FAOSTAT (2021) were collected to compare with faba bean outputs by LPJ-GUESS. As cropland soil characteristics information, data in the top layer (30cm) were derived from the GGCMI (Global Gridded Crop Model
Intercomparison) phase 3 soil input data set (Jägermeyr et al., in prep.). In general, although the total cropland cover in a grid cell could change annually over the course of the simulation, the relative fractions of each crop species within that cover fraction were held constant.

In terms of timing of N fertilizer application, a recent synthesis-analysis conducted by Mourtzinis et al. (2018) clearly indicates that the split N application at planting and early reproductive stage resulted in significantly greater soybean yields than a single application
method. Mineral N fertilizer was thus applied to legumes at the time of sowing ($DS$=0) and flowering ($DS$=1.0) with the same application rate, while all manure was added to soils at the time of sowing as a single application. Data sources for mineral N fertilizer and manure over the period 1901-2014 were derived from Ag-GRID (AgMIP GRIDded Crop Modeling Initiative, Elliott et al., 2015) and Zhang et al. (2017), respectively.

### 2.5 Statistical methods

In order to quantify the agreement between modelled and observed variables, the coefficient of determination (adjusted $R^2$), relative bias (RB, Eq. (16)), absolute bias (AB, Eq. (17)) and the root mean square error (RMSE, Eq. (18)) were computed:

$$RB = \frac{M_i - O_i}{O_i} \times 100\% \tag{16}$$

$$AB = \frac{|M_i - O_i|}{O_i} \times 100\% \tag{17}$$



$$RMSE = \sqrt{\frac{1}{n}\sum_{i=1}^{n}(M_i - O_i)^2} \qquad (18)$$

where $M_i$ and $O_i$ indicate modelled and observed values, n is the number of observations. To evaluate the fit of the interannual variability of modelled and reported yields on country level, the standard deviation (SD) and Pearson correlation coefficient (*r*, Eq. (19)) were calculated:

$$r = \frac{\sum_{i=1}^{n}(M_i - \overline{M})(O_i - \overline{O})}{\sqrt{\sum_{i=1}^{n}(M_i - \overline{M})^2 \sum_{i=1}^{n}(O_i - \overline{O})^2}} \qquad (19)$$

where $\overline{M}$ and $\overline{O}$ represent modelled and observed mean, n is the number of reported years.

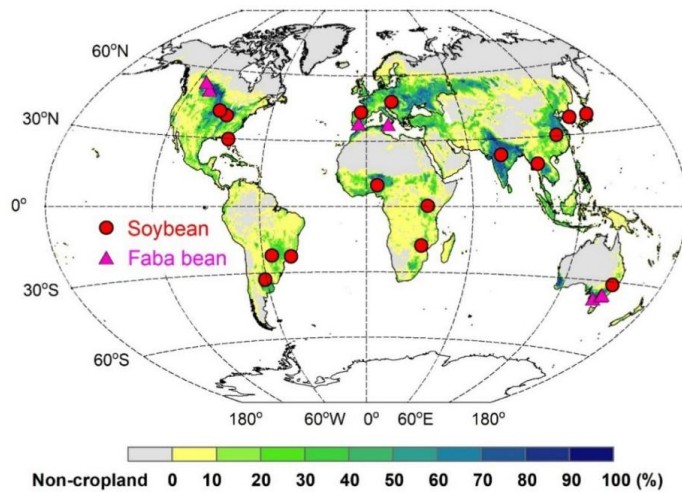

**Figure 3.** Spatial distribution of soybean (red circles) and faba bean (magenta triangles) sites used for BNF evaluation. The map background is cropland fraction (%) averaged over 1996-2005 at the resolution of 0.5° × 0.5°, derived from the LUH2 dataset (Hurtt et al., 2020)





# 3 Results

## 3.1 Model evaluation at site scale

### 3.1.1 Model performance across all sites

In order to examine model performance in simulating BNF-related variables across all grain legumes sites described in Table S2, we compiled six widely-measured variables related to N fixation at harvest shown in Fig. 4. Modelled yields generally agreed well with observations, especially in the site-specific simulation set-up. These had higher regression slope (0.83) and lower absolute bias (28%) compared with the global-uniform simulation set-up (Fig. 4a). N content in grains and shoots showed a lower agreement, with simulated values underestimating the observations for most sites (Fig. 4b-c), likely arising from two important N sources to grain legumes not being captured well by the model (i.e., soil N uptake and BNF, shown in Fig. 4d-e). The global-uniform run did not capture observed N fixation well, with a regression slope of 0.22 and absolute bias of 39%. The simulated BNF compared to observations was notably improved when using site-specific parameters, with the regression slope increasing to 0.41 and the absolute bias reducing to 31% (Fig. 4e). The field-based measurements showed that the N derived from the atmosphere (%Ndfa) was the main contributor to the legumes' total N uptake, ranging from 15 to 95%, with a mean of 64% across all field trials. LPJ-GUESS generally captured the mean response well, with simulated %Ndfa being 60% and 58% in the site-specific and global-uniform runs, respectively, despite several extreme disagreements at several faba bean sites (Fig. 4f).

A linear relationship between legume yields and the rate of BNF was found across a range of field sites in this study (Fig. S3a). Simulations from LPJ-GUESS mostly captured the close correlation between these variables, with $R^2$ ranging 0.46-0.63 ($p<0.001$) in both runs, not far from the measured value 0.67 (Fig. S3a). Linear regression parameters (i.e., slope and intercept) in both runs were close to the observations, indicating that the model reproduces well the N fixation effect on yield for individual sites.

A negative exponential relationship was observed between N-fertilizer application rate and N fixation across the field trials (Fig. S3b). LPJ-GUESS reasonably reproduced the decreased trend of BNF to N-fertilizer increase, with the similar fitting functions to observations, although higher N fixation rates were modelled in the highest-fertilized trial (600 kg N ha$^{-1}$) compared with measurements (Fig. S3b).

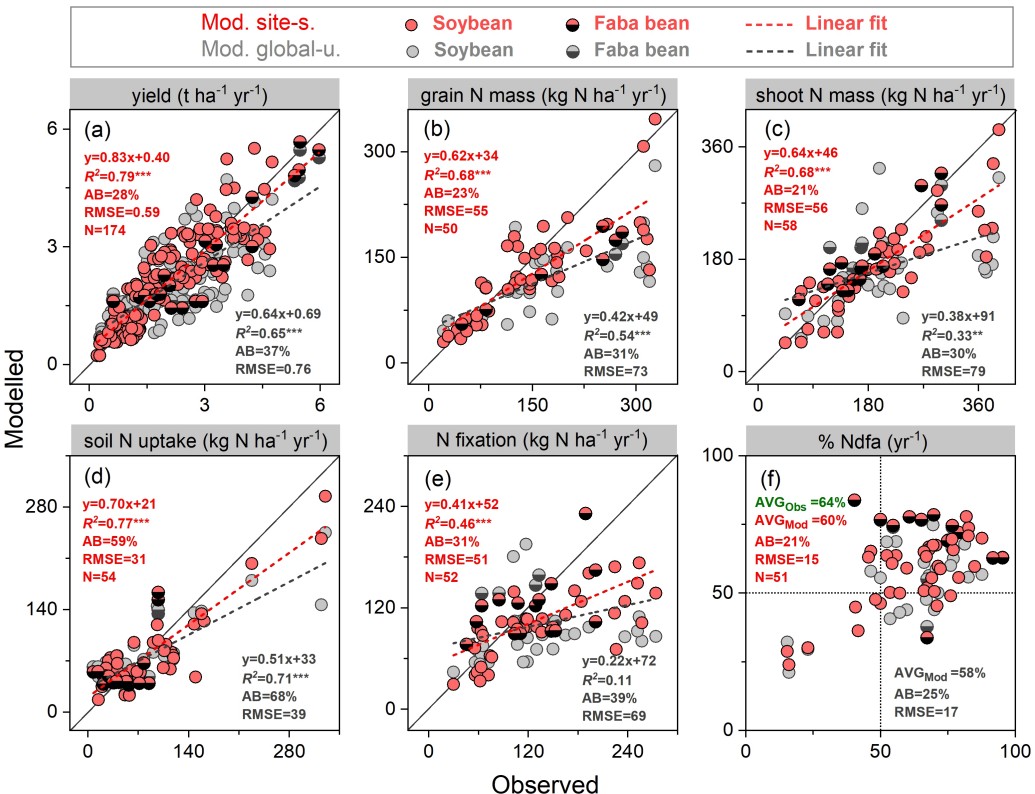

**Figure 4.** Comparison of modelled and observed yield (a), grain N mass (b), shoot N mass (c), soil N uptake (d), BNF (e) and %Ndfa (the proportion of plant N derived from the atmosphere) (f) at harvest across all soybean and faba bean sites. Filled red and grey circles depict 'site-specific' and 'global-uniform' run, respectively. The dashed line is fitted linear regression; *** and ** denote regressions statistically significant at p=0.001 and 0.01, respectively; AB is absolute bias (Eq. (17)), represented in percent (%); the unit of RMSE is the same as the associated variable; AVG in (f) is the averaged value of %Ndfa across all field trials.

### 3.1.2 Response to irrigation

The ability of the model to simulate the observed response of soybean tissues' biomass and N mass to irrigation management was examined using data from an experiment with rain-fed and irrigated treatments in Florida, U.S. (82.4°W, 29.6°N; see Table S2). Since the timing and quantity of irrigation was not reported in the literature (DeVries et al., 1989a; 1989b), we assumed that soybean was irrigated automatically when it experienced water stress in  the model, with the amount of plant water deficit as supplemental irrigation.

The mean observed grain yields at harvest were 2.0 and 2.9 t ha⁻¹ under rain-fed and irrigated conditions respectively, whereas the modelled yields were 1.9 and 2.5 t ha⁻¹ for the site-specific parameter run, and 1.6 and 2.1 t ha⁻¹ for the global-uniform parameter run, suggesting good model performance for rain-fed crops but an underestimation of the effect of irrigation on yields (Fig. 5a). Grain dry matter over the cropping season was simulated to increase by 32% and 45% on average in response to irrigation in the site-specific and global uniform runs, respectively. The observations show a similar response but with a higher increase of 75%. The modelled increase in grain N content caused by irrigation also showed a good agreement, with an increase of 35-58% in both runs, in line with the observed response of 42% (Fig. 5b).





The model generally reproduced observed leaf biomass and N mass better than the total above-ground production under rain-fed and irrigated treatments, with higher accuracy in the site-specific run. Over the growing season there was an obvious underestimation of the total above-ground production of biomass for both runs (Fig. 5a). This may be partly due to the fact that LPJ-GUESS at this point does not model soybean hulls, which account for ~15-20% of the total above-ground dry matter at harvest in the U.S. soybean rain-fed

cropping system (Córdova et al., 2020). The observed increase in shoot and leaf biomass due to water supply was 19% and 21%, respectively. In comparison, the site-specific parameterized model resulted in increases of 13% and 14%, respectively (15% and 14%for the global-uniform parameter run, see Fig. 5b). Overall, the observed soybean tissues' biomass and N content under rain-fed and irrigated conditions, and their response to irrigation management were captured reasonably well by the model at the U.S. Florida site, despite some deviations from observations in some cases.

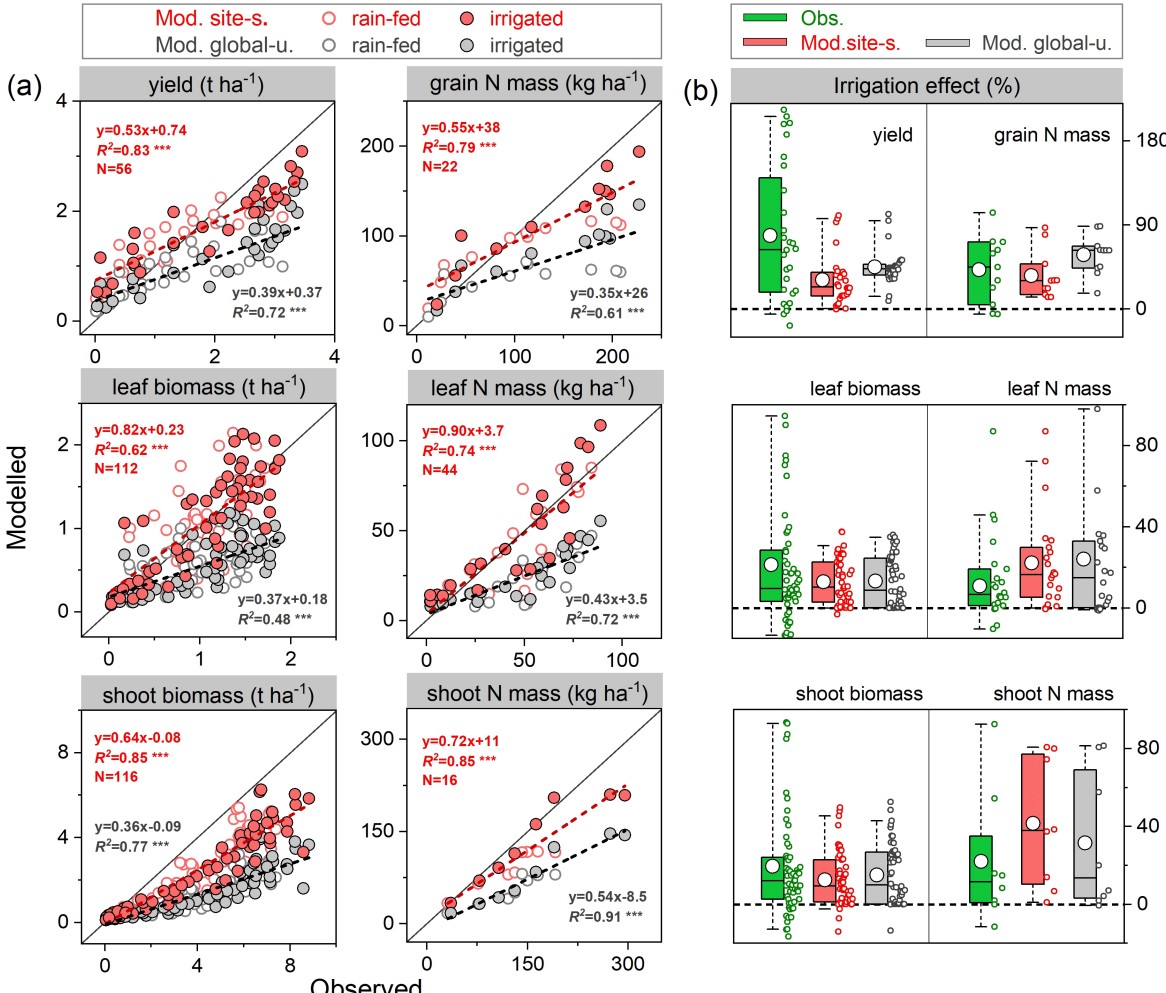


**Figure 5.** Comparison of modelled and observed soybean tissues' biomass and N mass (a), and their responses to irrigation management (b), comparing with those grown at rain-fed conditions. Red and grey circles depict 'site-specific' and 'global-uniform' run, respectively; The dashed line is fitted linear regression; *** denotes the regression statistically significant at p=0.001. Box plots in (b) denotes the 5th and 95th percentiles with whiskers, median and interquartile range with box lines, and mean with white dot (all data distributed next to the box). The seasonal data at each

phenological stage for tissues' biomass are available from 1978-79 and 1984-85 with rain-fed and irrigated treatments, those for N mass are achievable in 1979 and 1984, while the seasonal shoot N mass is only available in 1984.



### 3.1.3 Response to nodulating soybean

In Zapata et al. (1987), two field trials with non-nodulating and nodulating soybean were conducted in Seibersdorf, Austria (16.5°E,48.0°N, see Table S2), resulting in different plant C and N production at various growth stages. As described in Sect. 2.3,
nodulation process of legume has not yet been implemented in LPJ-GUESS, we thus switched off (on) the BNF function in the model to simply represent the non-nodulating (nodulating) soybean experiment.

During the growing season, yield, and grain N mass in the field trials increased rapidly after the vegetative stage, peaking around harvest. Simulations from LPJ-GUESS mostly captured those seasonal dynamics and the response to nodulating soybean (Fig. 6a-b): The modelled increase in yield and grain N mass due to nodulation was 34% and 51% in the site-specific run (34% and 45% in the
global-uniform run), respectively, in line with the observed response of 20% and 41% at harvest (Table 2), which suggests appropriate sensitivity of yield and N content in grain to N addition from N fixation. Similarly, the model generally reproduced the observed seasonal pattern of shoot N mass well, but with some underestimations in the nodulation trial (Fig. 6c).

Accumulated soil N uptake was captured reasonably well over the entire growing season, with higher accuracy at harvest in the global-uniform simulation (Fig. 6d). Measured mineral N uptake from soils declined on average by 25% in response to nodulation. In
comparison, the simulated reduction in uptake was 50% and 46% for the site-specific and global-uniform runs (Table 2). The BNF rates were low at the early growth stages when nodules were still establishing and increased rapidly between floral initiation and the early seed-filling, after which nodule senescence occurred and the increase in N fixation rate declined until physiological maturity (Fig. 6e). Simulations from LPJ-GUESS reproduced the seasonal pattern of N fixation with some overestimations in the accumulated BNF at the end of the growth period; the site-specific and global-uniform runs simulated 113 and 140 kg N ha$^{-1}$, respectively, compared to
the measured value of 103 kg N ha$^{-1}$ (Table 2).

**Table 2.** Comparison of modelled and observed yield (t ha$^{-1}$), grain N mass (kg N ha$^{-1}$), shoot N mass (kg N ha$^{-1}$), soil N uptake (kg N ha$^{-1}$) and N fixation rate (kg N ha$^{-1}$) from a soybean nodulation and non-nodulation experiment at harvest. The observed data were compiled using Tables 2- 4 in Zapata et al. (1987).

|  | Nodulation | | | Non-nodulation | | | Nodulation effect (%) | | |
| --- | --- | --- | --- | --- | --- | --- | --- | --- | --- |
|  | Obs. | Mod. site-s. | Mod. global-u. | Obs. | Mod. site-s. | Mod. global-u. | Obs. | Mod. site-s. | Mod. global-u. |
| Yield | 3.01 | 3.24 | 3.06 | 2.42 | 2.41 | 2.29 | 20 | 34 | 34 |
| Grain N mass | 162 | 166 | 148 | 115 | 110 | 102 | 41 | 51 | 45 |
| Shoot N mass | 222 | 198 | 181 | 158 | 134 | 138 | 41 | 48 | 31 |
| Soil N uptake | 119 | 76 | 86 | 158 | 152 | 159 | -25 | -50 | -46 |
| N fixation | 103 | 140 | 113 | - | - | - | - | - | - |



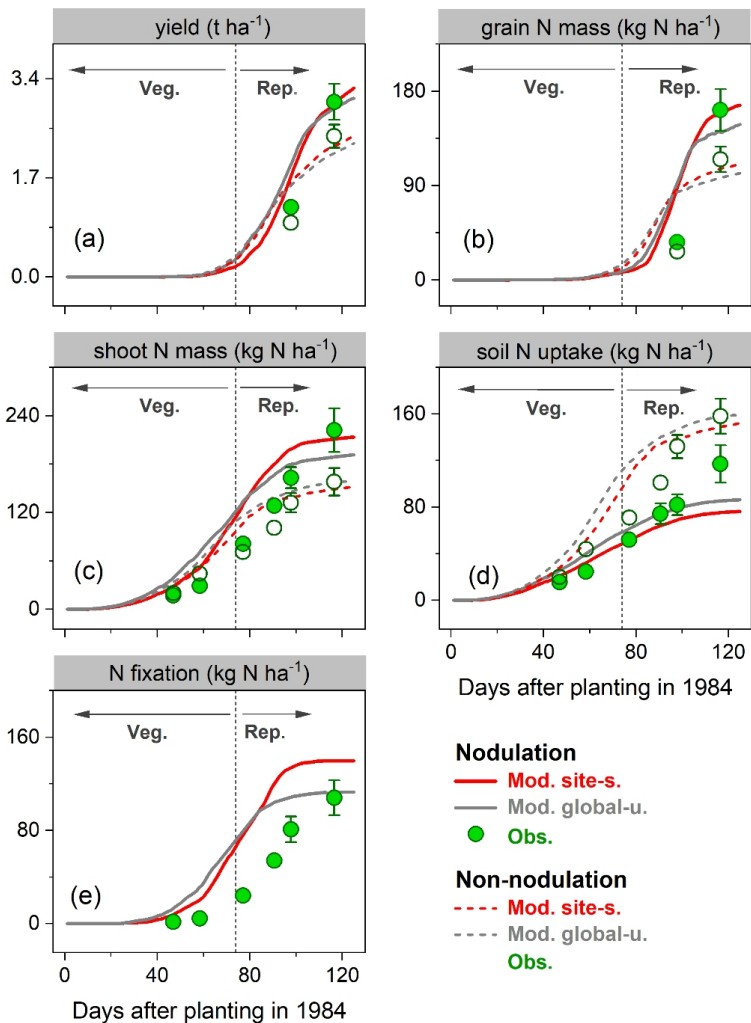


**Figure 6.** Observed (circles) and modelled (lines) yield (a), grain N mass (b), shoot N mass (c), soil N uptake (d) and BNF (e) for a field site in Austria (Zapata et al., 1987) for the cropping season 1984 with nodulating and non-nodulating soybean. The observed values of soil N uptake and BNF across all growth stages were calculated based on Figure 1 given in Zapata et al. (1987), and the vertical bars represent the standard error of four replicates' mean in original literature. Veg. and Rep. indicate vegetative and reproductive growth phase, respectively.

**3.1.4 Response to N-fertilizer in faba bean**

In the N-fertilizer experiment from Mínguez et al. (1993), four field trials were compared with N applications between 0 and 300 kg N ha$^{-1}$ at three crop growth stages and two faba bean varieties, grown in a Mediterranean climate (Spain, 4.8°W, 37.9°N, see Table S2). Over the entire growing season, leaf biomass and N content in the field trials increased until around May, after which leaf senescence started and biomass and N content declined (Fig. 7a-b). The model broadly reproduced these seasonal patterns, and the response to

different N application rates. The largest difference between modelled and measured leaf biomass was found at the end of the growing season, as a result of simulated leaf senescence rate being much lower than derived from measurements (Fig. 7a). In addition, the simulations showed modelled leaf N mass to decline rapidly during the late reproductive phase. This can be attributed to the transfer of N from vegetative parts to grain because of the high N demand in seeds during the grain filling period.



As seen in Fig. 7c, modelled soil N uptake was stimulated by soil mineral N availability, with an increase of 120-160% compared to
the unfertilized treatment. In contrast, fixing N from the atmosphere was constrained in the presence of elevated levels of soil mineral
N, with a reduction of 15-20%. The total N uptake for the cropping season 1987–88 was observed to only increase by 3% in response
to N application, as a consequence of the inoculation implemented in the unfertilized treatment (Mínguez et al., 1993). By contrast,
LPJ-GUESS produced relatively large increases of 14-16% in both runs, resulting in the observed increase in plant biomass and N
mass accumulation caused by N addition being largely overestimated in the model (Fig. 7c).

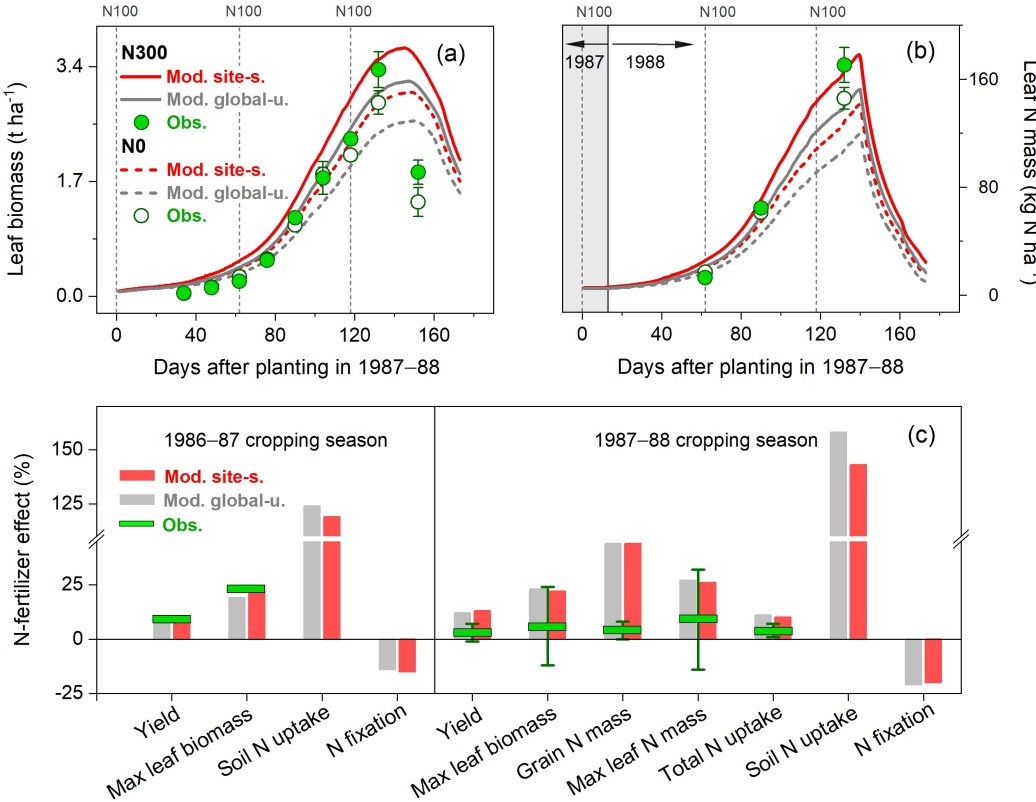


**Figure 7.** Observed and modelled seasonal pattern of leaf biomass (a) and leaf N (b) of faba bean in Spain for the cropping season 1987–88, with two
different levels of N-fertilizer input (0 and 300 kg N ha$^{-1}$ represented as N0 and N300, respectively), and response of faba bean yields and N uptake
to fertilized treatment at harvest (c), comparing with those grown at unfertilized conditions in the 1986–87 and 1987–88 cropping seasons. The
observed values were derived from the average of two faba bean varieties described in Mínguez et al. (1993), and their measured ranges are shown
by the vertical bars. The vertical dashed lines in (a)-(b) represent the timing and amount of fertilizer applied in the N300 treatment.

## 3.2 Model evaluation at global scale

### 3.2.1 Attained yields

Applying the global-uniform parameters described in Sect. 2.4.1, combined with the time-dependent gridded N-fertilizer data set
introduced in Sect. 2.4.2, we simulated soybean and all pulses (applying the faba-bean parameterization, see Sect. 2.4.2) at global
scale. We computed data for the period 1996-2005, since crop-specific fractional cover from the MIRCA data set was available for the
year 2000 (Portmann et al., 2010).





Modelled yields in the top ten soybean-producing countries showed a good agreement, with a higher $R^2$ of 0.52 (p<0.001) and lower RMSE value of 0.8 t $ha^{-1}$ $yr^{-1}$ when low-productivity countries (defined as all countries not belonging to the top ten producer countries) were excluded. With all producer countries included, $R^2$ of 0.17 (p<0.001) and RMSE of 1.4 t $ha^{-1}$ $yr^{-1}$ was found (Fig. 8a). LPJ-

GUESS generally tended to overestimate the reported yield for most countries where soybean production is low (e.g., most African countries, see Fig. 9a), with a mean relative bias in such countries of 81% (Fig. 8a). Modelled low yields were found in some arid and semi-arid countries (e.g., Egypt, Iran, and Turkey), with the underestimation spanning from 10-70% (Fig. 9a). Overestimated yields were also found when comparing simulated yields using the faba-bean parameterization against FAO reported values for pulses in general, with an overestimation also visible for some of the top producing countries (Fig. 8b). Likely the higher yields simulated by

LPJ-GUESS arise from the fairly high N fixation capacity simulated with the faba-bean parameterization (see Sect. 3.2.2), as well as the wide distribution of pulses worldwide, which grow under a broad range of climate and soil conditions.

A good fit of the interannual variability of modelled and reported yields is a further indicator of model performance. Despite the deviation between the model and observations for individual years, simulated variation in soybean yield over the period 1981-2016 matched well with reported yields among the top ten producer countries—especially in Argentina, India, and China—with a high

Pearson correlation coefficient (*r*) around 0.60 (p<0.001) and similar standard deviations (Fig. 9). The degree of yield variability between years was larger than seen in the FAO records, especially in the U.S., Canada, and Italy (Fig. 9), indicating high sensitivity of modelled soybean yield to changing environmental factors on spatial scales, such as weather, N-fertilizer application rates, and climate-related N fixation.

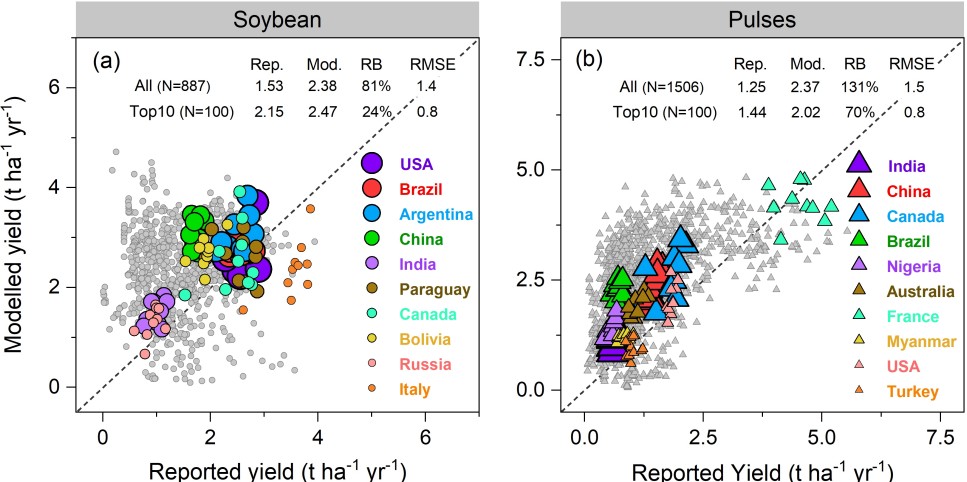

**Figure 8.** Per-country-and-year comparison of modelled yields of soybean (a) and pulses (b) against reported FAO statistics from 1996–2005. Each filled circle in (a) represents one year and one country; thus, a country can have up to 10 circles over 1996–2005. In total, 887 and 1506 country-year yield data points were used for comparison in soybean and pulses, respectively. The top 10 producer countries shown in color were chosen based on their total production over the same period, and marker size from large to small indicates their total relative production in descending order. Rep. and Mod. denote, respectively, reported and modelled yield (t $ha^{-1}$ $yr^{-1}$) averaged from 1996–2005. RB is relative bias (Eq. (16)), represented in percent

(%). The unit of RMSE is the same as yield (t $ha^{-1}$ $yr^{-1}$).



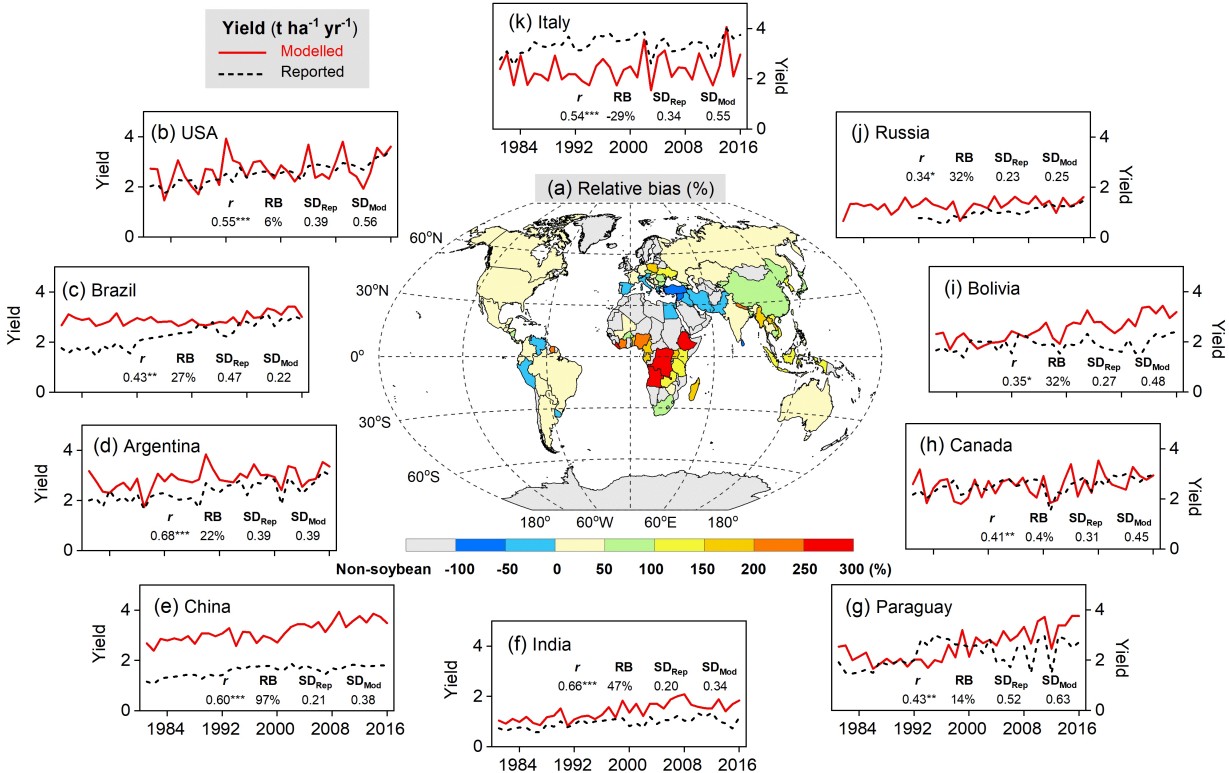

**Figure 9.** Comparison of simulated and FAO-reported yields on country level averaged over 1996–2005 (a), as well as time series of modelled soybean yield (red solid line) and reported FAO statistics (black dashed line) in the top 10 producer countries over the period 1981–2016. The top 10 producer countries (b–k, in descending order) were chosen based on their total production from 1996–2005. *r* is Pearson correlation coefficient (Eq. (19)), where ***, ** and * denote the correlation to be statistically significant at p=0.001, 0.01, and 0.05 level, respectively. RB is relative bias (Eq. (16)), represented in percent (%). $SD_{Rep}$ and $SD_{Mod}$ denote, respectively, reported and modelled yield standard deviations (t ha$^{-1}$ yr$^{-1}$) from 1981–2016.



### 3.2.2 N fixation and %Ndfa

The modelled spatial pattern of soybean N fixation showed large spatial variation (Fig. 10a). Modelled BNF rates as high as 250 kg N ha$^{-1}$ yr$^{-1}$ were found in western South America and most of Africa, where neither water nor temperature were critical limitations for N
fixation. Moreover, the relatively low fertilizer application in Africa leaves a nitrogen deficit that causes enhanced soybean N fixation. In contrast, in arid and semi-arid regions, soil water constrains BNF, while temperature limitation is seen in high latitudes and alpine areas (e.g., Andes in Peru). BNF rates in most regions (South Asia, West Asia, Sub-Saharan Africa and northwest China) were as low as 50 kg N ha$^{-1}$ yr$^{-1}$, particularly in Pakistan and northern India, where simulated BNF is severely constrained by the extreme high temperature over the cropping season. Eastern United States, Europe, Southern China and central-west Brazil showed intermediate
fixation rates, which were greater than 150 kg N ha$^{-1}$ yr$^{-1}$. Overall, the spatial variation of modelled legume BNF rate reflects to large degree the spatial climate patterns, in addition to N-fertilizer application. The low modelled %Ndfa of 45±3% in East Asia may reflect high N uptake from soils in response to substantial fertilizer investment in China over the past 40 years. In contrast, the modelled %Ndfa in Africa—with lower N application rates—was as high as 70±3%, although still lower than the reported mean value of 77% (Table 3). The spatial response of N fixation rate to climate constraining factors (i.e., soil temperature and water) is shown for
pulses in Fig. S4.

At regional scale, the modelled outputs compare well with N fixation rates from the literature (Fig. 10b-f, Table 3). For example, in South America and North America, both major soybean-producing regions, simulated BNF rates were 156±14 and 127±44 kg N ha$^{-1}$ yr$^{-1}$ over the period 1981-2016, respectively, compared with literature-derived values of 136 and 144 kg N ha$^{-1}$ yr$^{-1}$ (Peoples et al., 2009). Globally, the modelled soybean N fixation rate of 132±21 kg N ha$^{-1}$ yr$^{-1}$ was reasonably consistent with the meta-analysis result
of 111-125 kg N ha$^{-1}$ yr$^{-1}$ in Salvagiotti et al. (2008) and the FAO-based estimate of 176 kg N ha$^{-1}$ yr$^{-1}$ from Herridge et al. (2008). The contribution of N fixation to total N uptake in soybean was somewhat underestimated in several regions. A similar trend to underestimate reported %Ndfa was also found for pulses (Table 3).

Having large soybean planting areas and high yields, South America and North America contributed 80% of simulated global soybean N fixation, followed by East Asia, South Asia and Europe (Table 3). Globally, simulated annual N fixed over the period 1981-2016
was 11.6±2.2 Tg in soybean, which showed a good agreement with the estimate of 16.4 Tg N reported by Herridge et al. (2008) and the extrapolated result of 10.4 Tg N estimated by Gelfand and Robertson (2015) based on U.S. field trials. However, we modelled pulses to fix 5.6±1.0 Tg N annually, almost two times higher than the 2.95 Tg N estimated by Herridge et al. (2008). The difference in the case of pulses is most likely due to the low N fixation rate used by Herridge et al, ranging from 23-107 kg N ha$^{-1}$ yr$^{-1}$, lower than the mean value of 119±15 kg N ha$^{-1}$ yr$^{-1}$ modelled by LPJ-GUESS (Table 3).



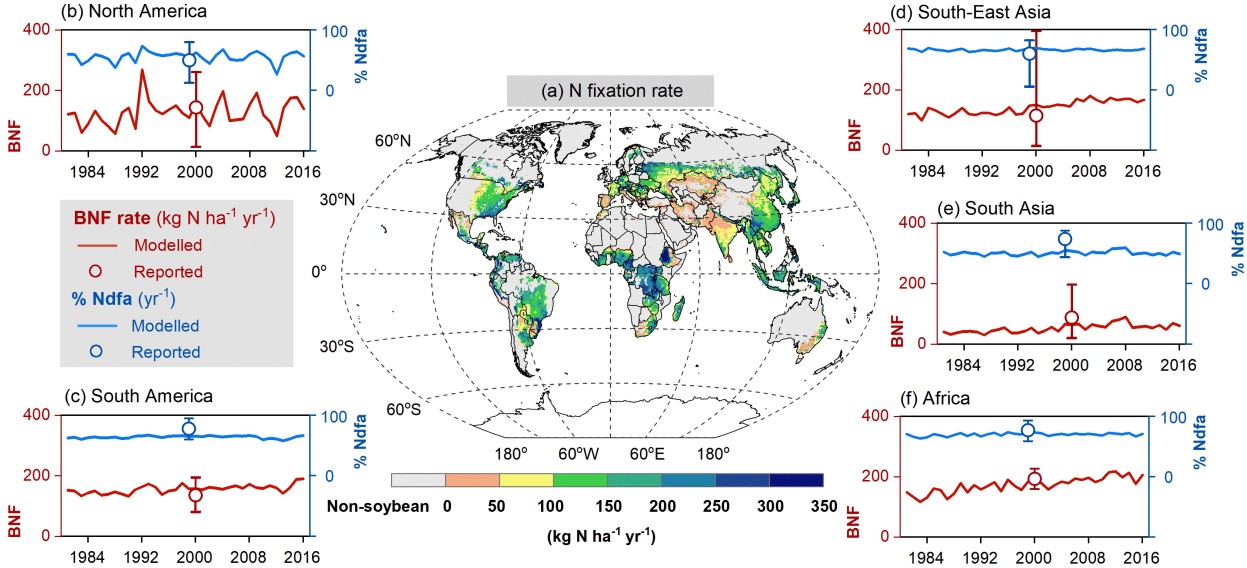


**Figure 10.** Map of soybean N fixation modelled by LPJ-GUESS averaged over 1996–2005 (a), and the comparison of simulated BNF rate (red line) and %Ndfa (blue line) with literature-reviewed data (open circle; Peoples et al., 2009) on regional level (b–f). Reported data shown in open circles do not represent specific years but the potential over time in Peoples et al. (2009), the vertical bars denote the range of estimations based on original literature given in Table 1 in Peoples et al. (2009).





**Table 3.** Modelled continent-level biological N fixation rate, the proportion of plant N derived from the atmosphere (%Ndfa), and total N fixation in soybean and pulses for the time period 1981–2016, compared to estimates from the literature with the reported range in brackets. The modelled results are represented as mean ± 1 standard deviation.

| | Soybean | | | | | Pulses | | | | |
|---|---|---|---|---|---|---|---|---|---|---|
| | N fixation rate (kg N ha⁻¹ yr⁻¹) | | %Ndfa ( yr⁻¹) | | Total N fixation (Tg N yr⁻¹) | N fixation rate (kg N ha⁻¹ yr⁻¹) | | %Ndfa ( yr⁻¹) | | Total N fixation (Tg N yr⁻¹) |
| | Reported | Modelled | Reported | Modelled | Modelled | Reported | Modelled | Reported | Modelled | Modelled |
| **South Asia** | 88 [a] (21-197) | 53±14 | 74 [a] (44-88) | 51±3 | 0.4±0.1 | - | 62±12 | - | 52±2 | 0.9±0.2 |
| **South-East Asia** | 115 [a] (0-400) | 141±22 | 60 [a] (0-82) | 66±2 | 0.2±0.0 | - | 139±16 | - | 69±1 | 0.3±0.1 |
| **Africa** | 193 [a] (159-227) | 172±25 | 77 [a] (65-89) | 70±3 | 0.2±0.1 | - | 157±21 | - | 70±1 | 1.9±0.5 |
| **North America** | 144 [a] (14-311) | 127±44 | 50 [a] (13-80) | 56±9 | 4.9±1.7 | 118 [b] (13-252) | 137±21 | 74 [b] (60-92) | 59±3 | 0.6±0.1 |
| **South America** | 136 [a] (80-193) | 156±14 | 78 [a] (60-95) | 64±2 | 4.5±1.1 | - | 157±18 | - | 66±3 | 0.5±0.1 |
| **East Asia** | - | 101±16 | - | 45±3 | 1.2±0.2 | - | 114±17 | - | 49±4 | 0.4±0.1 |
| **Central Asia** | - | 63±19 | - | 36±6 | 0.0±0.0 [e] | - | 104±21 | - | 59±4 | 0.0±0.0 [e] |
| **West Asia** | - | 27±7 | - | 14±3 | 0.0±0.0 [e] | 100 [b] (78-133) | 65±10 | 69 [b] (63-76) | 35±4 | 0.2±0.0 |
| **Europe** | - | 117±17 | - | 54±4 | 0.2±0.1 | 153 [b] (73-211) | 177±26 | 74 [b] (60-92) | 63±4 | 0.6±0.1 |
| **Oceania** | - | 78±27 | - | 38±9 | 0.0±0.0 [e] | 143 [b] (82-216) | 126±23 | 82 [b] (69-89) | 37±6 | 0.4±0.1 |
| **Global** | 111-176 [a, c, d] | 132±21 | 52-68 [a, c, d] | 57±4 | 11.6±2.2 | 107-129 [b, d] | 119±15 | 75 [b, d] | 60±1 | 5.6±1.0 |

[a] Soybean data in Peoples et al. (2009); [b] Faba bean data in Peoples et al. (2009); [c] Salvagiotti et al. (2008); [d] Herridge et al. (2008); [e] the values do not represent zero grain legume planting area in that region.



## 4 Discussion

### 4.1 Model performance at site scale

The overall model agreement with measured legume yield and grain N mass was good across a range of field sites (Fig. 4). Values at harvest were on average about 20-30% lower than values reported in the measurements (Fig. 4a-b). A similar, small, underestimation was found in the shoot N mass (Fig. 4c), indicating that the productivity generally is somewhat too low in the model. One factor contributing to the underestimation is that LPJ-GUESS applies a conversion factor of 2.0 from plant C mass to dry matter (Smith et al., 2014), ~10% lower than the published measurement of 2.24 reported in Osaki (1993). In addition, we found that the model underestimated above-ground biomass while simultaneously overestimating below-ground productivity at the three sites where measured root biomass was available. This could be addressed by adjusting the root:shoot allocation (i.e., modifying the daily assimilate partitioning function in grain legumes; Eq. (5)), but this is currently prevented by the lack of sufficient observed root biomass information.

Modelled soil N uptake was sensitive to soil mineral N concentration and hence driven by fertilizer application rates (Fig. 7c; Fig. S3c). Generally, LPJ-GUESS tended to overestimate soil N uptake in regions where legumes were not or only little fertilized (Fig. S3c). This might be partially due to the selected legume cultivars at the experimental plots, which have been reported to have low mineral N uptake potential (Gan et al., 2002, 2003; Santachiara et al., 2017, 2018). Moreover, the saturation effect of mineral N concentration on N uptake implemented in the model might result in the discontinuation of N uptake when soil available N is abundant (Zaehle and Friend, 2010; Warlind et al., 2014). Under high fertilization rates (up to 260-600 kg N ha$^{-1}$, Fig. S3c), a strong underestimation in soil N uptake was expected because of the modelled saturation-response to high soil mineral N, resulting in little changes in the level of soil N uptake no matter how much N-fertilizer was applied.

Adding mineral N to the soil in LPJ-GUESS can increase soil N uptake, reducing the plant's N deficit and therefore also reducing the upper limit of daily N fixation rate (Fig. 2). Although the modelled negative relationship between fertilizer application rates and N fixation showed a generally good agreement with the observed response across a range of field sites, the simulated BNF rates at the high-fertilized trials (i.e., 260-600 kg N ha$^{-1}$, Fig. S3b) were about 50-80% higher than the measured values (Fig. 4e; Fig. S3b). This might be partially explained by the underestimation in soil N uptake under higher N concentration, resulting in plant N demand remaining very high, and substantial N still being fixed. The large discrepancies between modelled and observed N uptake in the high-fertilized treatments suggest that the N uptake representation in LPJ-GUESS should be further improved. A step forward could be to incorporate the inhibitory effects of soil mineral N content on N fixation into the model (Chen et al., 2016; Wu et al., 2020), since experimental evidence indicates that high soil mineral N not only affects plant N uptake in roots, but also depresses legume nodule initiation, nodule size and specific nodule activity, therefore reducing the amount of N fixation from the atmosphere (Herridge et al., 1984; Purcell and Sinclair, 1990; Thornley and Cannell, 2000).

The percentage of plant N derived from the atmosphere (i.e., %Ndfa) is a key parameter required for quantifying N fixation in the field and varies widely, caused by differences in climate, soil type and degree of N fertilization (Herridge et al., 2008). LPJ-GUESS captured the range and mean value of %Ndfa well across different field trials, with some disagreements, especially for faba bean (Fig. 4f). An underestimated %Ndfa is likely caused by the combined effects of underestimated N fixation (Fig. 4e) and overestimated soil N uptake (Fig. 4d). Nevertheless, we found modelled %Ndfa to decline with increasing N fertilizer application, which is also the observed response in the field trials. A negative correlation between %Ndfa and fertilizer application rates was also reported by Salvagiotti et al. (2008). These results all suggest that LPJ-GUESS is able to effectively capture the observed overall patterns of soil mineral N uptake and N fixation in grain legumes and their responses.





Since the SLA and C:N ratio of plant organs play a vital role in determining N uptake when modelling vegetation C-N dynamics (Olin et al., 2015), it is to be expected that applying measured values for site-scale modelling resulted in much better agreement when
comparing simulation results to measurements (Figs. 4-7). Remaining discrepancies between modelled and observed N-cycle variables may reflect missing processes in the model, such as inoculation effectiveness, phosphorus limitation, and soil acidity, especially in terms of inoculant application. Field experiments have shown that proper inoculation of rhizobia promotes nodulation and results in an efficient increase in N fixation, although large variations exist between strains of rhizobia (Mínguez et al., 1993; Sanginga et al., 1997; Tewari et al., 2004; Denton et al., 2017). Using a fixed parameter ($N_{maxfixpot}$, Eq. (9)) to represent all inoculation situations such as in a
global uniform calibration cannot reflect this variability. In addition, due to the difficulties in measuring both nodules and roots in the field directly, in many studies the observed BNF rates were determined from plant above-ground biomass. Excluding the root contribution to the whole plant BNF rates most likely result in an underestimation of N fixation (Córdova et al., 2019, 2020): N associated with nodules and roots in soybean and faba bean may account for 20-40% of the total N accumulation at mid-flowering phase (Unkovich and Pate, 2000; Khan et al., 2003).

Compared to non-BNF (i.e., non-nodulation treatment, see Sect. 3.1.3), BNF in LPJ-GUESS greatly improves simulated soybean yield and aboveground N mass, with an overall increase in both variables of 30-50% (Table 2). Córdova et al. (2019) found a yield increase of 150% in response to nodulation in an unfertilized treatment, but that increase reduced to 55%—similar to our modelled yield increase—when a high N input was applied (i.e., 135 kg N ha$^{-1}$). N fixation can help grain legumes to dramatically enhance their total N accumulation and to achieve higher N concentration in seeds; however, these benefits are accompanied by the increase in
respiration cost of 4-16% of fixed total photosynthetic carbon (Kaschuk et al., 2009, 2010). Such a respiratory photosynthate consumption would reduce productivity if photosynthesis rate does not increase to compensate for the cost. In LPJ-GUESS, as described in Sect. 2.3, we assumed that up to 50% of daily NPP can be consumed to fix N. This approach has the advantage that legumes are able to maximize photosynthetic gain due to reduced N limitation in carboxylation capacity ($V_{max}$), but it entails the risk of lower productivity if too much NPP is invested into fixation. Nevertheless, we did not observe any modelled extreme C expense on
N fixation over the entire growing season; soybeans were only spending up to about 40% of modelled daily NPP on fixation from site scale (Fig. S5) to a larger region (Fig. S6). Such NPP consumption was not only lower than our assumed upper limit of 50%, but also appropriately consistent with the reported range of 14-32% described by Kaschuk et al. (2009), demonstrating the reasonable C cost scheme implemented for N fixation in our model. Taken together, the modelled C profits due to N fixation can be attributed to the positive feedback between BNF and photosynthesis in LPJ-GUESS: C cost-based N fixation results in a higher rate of photosynthesis
because of the enhanced leaf N concentration; in turn, the increased rate compensates for the C cost, and allocates more assimilate to roots and thus enhances N fixation.

### 4.2 Global yields, N fixation and %Ndfa

Agreement between FAO-reported and simulated yields at country level was reasonable for the major soybean-producing countries. However, in some arid and semi-arid countries, the modelled yields were up to 70% lower than FAO-reported values probably
because of the simulated low N fixation rate caused by severe water constraints (Fig. S4). By contrast, LPJ-GUESS produced an overestimation of 100-300% in yield production among some African countries, with BNF rates of 300-350 kg N ha$^{-1}$ yr$^{-1}$ being modelled in these regions (Fig. 10a). More recent studies that report data from African farms have indicated that the soybean N fixation rate can be as low as 0-50 kg N ha$^{-1}$ yr$^{-1}$ in most farmers' fields, largely because of the inconsistent effectiveness of inoculation in the acid soils (Ulzen et al., 2016; Muleta et al., 2017; Vanlauwe et al., 2019). The BNF implementation and soil
representation in LPJ-GUESS do not account for inoculation effectiveness in response to soil pH.





In our simulations, the annual amount of N fixed by global grain legumes (i.e., soybean and all pulses) of 17.2±2.9 Tg averaged over the period 1981-2016 agreed well with the estimate of 19.4 Tg provided by Herridge et al. (2008), who used crop production statistics from FAOSTAT and legume-specific %Ndfa from farmers' fields for estimating global N fixation. In an earlier study, a total of 10 Tg N (range of 8-12 Tg N) was estimated from legume crops' BNF annually (Smil, 1999), far lower than our findings. The discrepancy
between the estimates in Smil (1999) and Herridge et al. (2008) likely reflect the lower values of % Ndfa for soybean and pulses used for calculations in Smil (1999). Also, Smil (1999) excluded below-ground fixed N associated with roots and nodules, which contributes to the low estimate. Our modelled N fixation from grain legumes amounts to ~12% of the annual mean of ca. 140 Tg N that were estimated to be fixed in all global terrestrial ecosystems (Cleveland et al., 1999, 2013; Galloway et al., 2004; Wang and Houlton, 2009; Vitousek et al., 2013; Meyerholt et al., 2016; Xu-Ri and Prentice, 2017; Yu and Zhuang, 2020; Davies-Barnard and
Friedlingstein, 2020), indicating the importance of BNF input in agricultural systems for the global terrestrial N cycle, although a large proportion of the fixed N is removed in grains from the ecosystems each year.

Currently, three environmental factors, soil temperature, moisture, and soil mineral N concentration, affect modelled N fixation. As discussed in Sect. 4.1, increased soil N availability would depress N fixation as plant total N can be met more 'cheaply' via soil mineral N uptake. This effect can be also seen from the spatial pattern of %Ndfa in the northern temperate region, such as the United
States, western Europe, and China. Here, anthropogenic N deposition, together with the intensive application of fertilizers result in soils being N-rich, inhibiting simulated BNF. This could explain why our modelled soybean N fixation rate was not high in East Asia and only contributed to 45±3% of plants' total N uptake (Table 3). In comparison, the high rate of N fixation found in tropical regions is primarily due to their high nitrogenase activity under warm and moist soil conditions (Fig. S4), resulting in %Ndfa of ~70% being modelled for all grain legumes in the tropics (i.e., Africa and South-East Asia; Table3). A similar spatial variation between temperate
and tropical regions in N fixation was also reported by other modelling studies in global terrestrial ecosystems (e.g., Wang and Houlton, 2009; Meyerholt et al., 2016; Xu-Ri and Prentice, 2017; Yu and Zhuang, 2020). Taken together, these results reveal that LPJ-GUESS broadly captures how N management practices and climate variation affect soil N uptake and biological N fixation in grain legumes at large spatial scales.

### 4.3 Modelling challenges and future work

Similar to most ecosystem and crop models, specific leaf area (SLA) in LPJ-GUESS is used to compute LAI and indirectly affects the amount of photosynthesis. SLA also further impacts plant total N uptake since the N demand in plant organs is always associated with the photosynthetic assimilate in the model. The disagreements between modelled and observed C-N variables on seasonal pattern (Figs. 6-7) can therefore be partially attributed to the static value of SLA implemented in LPJ-GUESS. Some studies have shown that SLA varies with crop growth development (Boote et al., 2002; Ainsworth et al., 2007) and environmental conditions (Poorter et al.,
2009). In addition, low temperature, excess radiation, water deficit, or rising $CO_2$ concentration would also result in reduced SLA through affecting leaf area expansion and internode elongation (Ainsworth and Long, 2005; Yin and Struik, 2010). Applying SLA as a constant in the model (see Sect. 2.4.1) cannot reflect these responses. Incorporation of dynamic SLA over the crop growing season and its response to the environment remains to be taken into account in future model development.

Despite many experimental studies on the limitation of soil water deficit on biological N fixation, the nature of the relationship
between legume BNF and soil water content is not well-characterized in models. A linear water-limitation function incorporated in LPJ-GUESS (Eq. (11)) implies, for instance, that the model has little potential to represent the situation when plants experience stress from excessive water (flooding). The impact of excess soil water on legume N fixation is either omitted or oversimplified in most crop models. For instance, a simple assumption adopted in Sinclair's model is that the N fixation process is stopped forcibly when flooding takes place (Sinclair et al., 1987). In STICS, the N fixation inhibition by water excess is represented as a stress from hypoxia in the



roots (Brisson et al., 2003). The process of legume BNF restraint by flooding is implemented into CROPGRO (Boote et al., 2008) by calculating the proportion of water-filled pore space. N fixation is assumed to be only restricted when all pore space is filled with water; however, this rule has not been well-evaluated so far.

Although high soil mineral N concentration suppresses legumes' root nodulation and further impacts N fixation (Xia et al., 2017; Mourtzinis et al., 2018; Brar and Lawley, 2020), a moderate level of soil N in vegetative growth stage is conducive to root growth and

nodule formation, stimulating N fixation (Waterer and Vessey, 1993; Salvagiotti et al., 2008). In the field trials a specific threshold of soil N concentration above (below) which N fixation is inhibited (stimulated) is hard to measure. In addition, the timing of N application remains a challenge (Córdova et al., 2020). Some studies reported that applying N fertilizer at planting as starter N can increase yield gains because of sufficient soil available N to stimulate early season soybean growth (Pikul et al., 2001; Osborne and Riedell, 2011; Gai et al., 2017). However, other studies argued that the best time to apply additional N would be expected at early

reproductive growth stages, during which legumes have the greatest N demand for seed development, also soil N reserves are depleting and N fixation rate starts slowing down (Mourtzinis et al., 2018; Córdova et al., 2019; Zhou et al., 2019). Unfortunately, as mentioned earlier, there are no consistent results on these measured factors, resulting in the difficulties in incorporating the mechanistic processes or setups into LPJ-GUESS at this point.

Taken together, the challenge of modelling legume N fixation is primarily due to its large variance between species and sites, and

managements. Symbiotic nitrogen fixation by rhizobia is an extremely complex natural process, which is associated not only with host plant's and soil N status in the macro-environment (see Fig. 2), but also with the process of *Rhizobium* or *Bradyrhizobium* bacteria in root nodules in the micro-environment (Rice et al., 2000). It is difficult to incorporate these two different but highly related processes into one model (Liu et al., 2011; Chen et al., 2016). Furthermore, there is inadequate amount of information available to establish a reliable relationship between BNF and other factors such as soil pH (Rice et al., 2000; Vanlauwe et al., 2019), inoculation

effectiveness (Tewari et al., 2004; Denton et al., 2017; Liu et al., 2019), salinity (Zahran, 1999; Bruning and Rozema, 2013) and other nutrition availability (Le Roux et al., 2009; Singh et al., 2012), which are currently missing from LPJ-GUESS and other crop models despite many field experiments demonstrating their importance.

### 5. Conclusions

In this study we implemented a mechanistic process of symbiotic biological N fixation in grain legumes into the crop module of LPJ-

GUESS. The modelled C-N variables of soybean and faba bean were extensively evaluated with observed data from site scale to a larger region. Our results showed that the BNF scheme adopted in LPJ-GUESS realistically responded to water and N managements, as well as to climate variation, and produced N fixation and yields which generally agreed with measurements.

Our model estimated that global biological N fixation in grain legumes (i.e., soybean and all pulses) was $17.2\pm2.9$ Tg N $yr^{-1}$ during the period 1981-2016 and that the highest fixation rate occurred in tropical and temperate regions with warm and moist climate. Soil water

and temperature were dominant controls on N fixation, in addition to N-fertilizer application rate. Processes missing from the model, such as inoculation effectiveness and soil acidity, might have biased our estimates on N fixation and yields at global scale.

The N dynamic process of N fixation with a C-N allocation scheme for crops in LPJ-GUESS provides an opportunity to estimate the changes in global grain legumes' production and global terrestrial C and N pools under future land-use or climate change scenarios. It can also help to predict and detect the potential contribution of N-fixing plants as "green manure" to reducing or removing the use of

N fertilizer in global agricultural systems, considering different climate conditions, management practices, and land-use change scenarios.

**Code and data availability**

Global daily climate data of GSWP3-W5E5 is available at https://www.isimip.org/gettingstarted/input-data-bias-correction/details/80/ (last access: 14 July 2021). National soybean and pulses yield statistics from FAOSTAT presented in this paper can be retrieved from

http://www.fao.org/faostat/en/#data/QC (in this study, last access: 9 May 2021). The rest of model input data and measurement results used in this study can be accessed at https://doi.org/10.5281/zenodo.5148255 (Ma et al., 2021).

LPJ-GUESS is worldwide tested, refined and developed, but the model code is managed and maintained by the Department of Physical Geography and Ecosystem Science, Lund University, Sweden. The source code can be made available with a collaboration agreement under the acceptance of certain conditions. The code used in this paper is available to the editor and reviewers via a

restricted link on condition that the code is only for review purposes, it has to be deleted after the review process. Additional details and information can be achieved at the LPJ-GUESS website (http://web.nateko.lu.se/lpj-guess, last access: 14 July 2021) or by contacting the corresponding author.

**Supplement**

The supplement related to this article is available online at XXX

**Author contributions**

AA, SO and JM conceived this study. SO and JM developed the model code. JM carried out the analysis and produced the figures. SO, PA and SSR assisted with data collection and parameter tuning. SSN provided soybean data in Kenya for model evaluation. JM wrote the original draft, with further editing from AA, SSR, ADB, PA, SO and SSN.

**Competing interests**

The authors declare that they have no conflict of interest.

**Acknowledgements**

We would like to thank Dr. Stijn Hantson for his technical support at the beginning of the model development. This research has been supported by German Federal Ministry for Economic Cooperation and Development (BMZ) and administered through the Deutsche Gesellschaft für Internationale Zusammenarbeit (GIZ) Fund for International Agricultural Research (FIA), grant number 81206681.

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
