# Peer review of "Modelling symbiotic biological nitrogen fixation in grain legumes globally by LPJ-GUESS (v4.0, r10285)"

_Geoscientific Model Development, 2021_

## Author Response (AR1)

**Response to executive editor Dr. Astrid Kerkweg** (Comments in black, Answers in blue, Revisions in red)

We would like to thank Astrid Kerkweg, the Executive editor of GMD, for her constructive comments on our manuscript regarding to the basic requirements of papers submitted in GMD. Our point-by-point response to the comments is given in the following:

*Dear authors,*

*in my role as Executive editor of GMD, I would like to bring to your attention our Editorial version 1.2: https://www.geosci-model-dev.net/12/2215/2019/*

*This highlights some requirements of papers published in GMD, which is also available on the GMD website in the 'Manuscript Types' section: http://www.geoscientific-model-development.net/submission/manuscript_types.html*
*In particular, please note that for your paper, the following requirement has not been met in the Discussions paper:*

Thank you for the notes. We have carefully read the information of Editorial version 1.2. We will follow the requirements of the papers published in GMD.

- *"The main paper must give the model name and version number (or other unique identifier) in the title."*

*Please add a version number for LPG-GUESS to the title upon your revised submission to GMD.*

Thank you. We have changed the title accordingly from "Modelling symbiotic biological nitrogen fixation in grain legumes globally by LPJ-GUESS " to "Modelling symbiotic biological nitrogen fixation in grain legumes globally by LPJ-GUESS (v4.0, r10285)" in the revised submission.

*Yours,*

*Astrid Kerkweg*

**Response to anonymous referee #1** (Comments in black, Answers in blue, Revisions in red)

**General comments:**

*The biological nitrogen fixing (BNF) in legumes is an important biological and chemical process in ecosystems. The paper under review was high capacity scientific load and added two legume crops with BNF capacity into a DGVM model LPJ-GUESS to simulate global nitrogen fixation in legume-based cropping systems. The topic of this manuscript is very interesting and the BNF calculation exercise is probably a worthwhile contribution to other N general models. Readers are glad to see such a progress in modelling N fixing crops in the DGVM models community (to my knowledge, an empirical function between BNF and ET (or NPP) is often used in some DGVM models to estimate the annual N fixation across all terrestrial ecosystems). Additionally, just for curiosity, I checked all the observed data authors used for site-scale evaluation and simulation (10.5281/zenodo.5148255), which matches with the original literature listed in the text and provides an opportunity for other models' evaluation. Overall, this manuscript is well-written and easy to follow, with reasonable BNF representation and solid evaluation results, I thus suggest to publish it in GMD after minor revision. However, a few of minor issues that need to be further addressed concerning assumptions made, methodology, and clarity of presentation.*

We thank the reviewer for the expressed interest in our BNF implementation methodology and the manuscript. In the revisions to the manuscript we will be addressing the raised questions as described below.

*Authors mentioned that N fixation is a high energy consumption process, with ~4-16% fixed C cost. But why gave the assumption of up to 50% daily NPP used for BNF in LPJ-G? I fully understand the purpose of setting up MAXbnfcost, in case of extreme cases taking place when modelling N fixation, but why the cost proportion is 50%, rather than 20% or 30%? Obviously, the value of 20% is much more close to the measured upper bound of 16%, no? Furthermore, carbon cost per unit fixed N (in the manuscript a fixed value of 6 is used in two legumes) is varying between crop growth stages (literally high C consuming at soybean V4-R2 stages), have you taken this into account?*

Starting with the first question: according to Kaschuk et al. (2009), the C cost for BNF in grain legumes is around 4-16%. This energy consumption refers to the fixed total photosynthetic carbon (i.e., GPP) by plants, which is approximately equal to 8-32% of NPP cost in the model (simply assuming GPP=2×NPP; Waring et al., 1998). Accordingly, the measured upper bound of NPP cost should be 32%, rather than 16%. Also, from the global map, our modelled NPP cost in most grid cells is ranging from 5-25% for both grain legumes (see Figs. S7-S8), therefore corresponding to the reported range of 8-32%. Only in a few cases, NPP cost is in fact greater than 50% in the model simulations. To reduce the effect of the $MAX_{bnfcost}$ constraints on BNF, we thus set up the upper limit as 50% to catch the few cases where it well exceeds the reported range with a realistic value. In the revised manuscript, we clarified the range of C costs in the Discussion, as well as in the revised explanation of Fig. S7.

Revision in Sect. 4.1: "In LPJ-GUESS, as described in Sect. 2.3, we assumed that up to 50% of daily NPP can be consumed to fix N. This approach has the advantage that legumes are able to maximize photosynthetic gain due to reduced N limitation in carboxylation capacity ($V_{max}$), but it entails the risk of lower productivity if too much NPP is invested into fixation. Nevertheless, in most cases our modelled NPP cost over the soybean growing season was ranging from 1-40 % at site scale (Fig. S7) and 5-25% on a large region (Fig. S8). Such NPP consumption was not only lower than our assumed upper limit of 50%, but also appropriately consistent with the reported range of 14-32% described by Kaschuk et al. (2009), demonstrating that the C cost scheme implemented for N fixation in our model is reasonable."

Revision in caption of Fig. S7: "Modelled BNF-limiting factors (a) and daily NPP for N fixation cost (b) in nodulating soybean treatment in the 1984 cropping season at an Austrian site (Zapata et al., 1987). Veg. and Rep. indicate vegetative and reproductive growth phase, respectively. The reported GPP cost range of N fixation (4-16%) was extracted from Table 1 given in Kaschuk et al. (2009), and converted to NPP cost (8-32%) by multiplying by 2.0, assuming GPP=2×NPP (Waring et al., 1998). "

The reviewer is correct that in principle high C cost for BNF basically takes place at soybean V4-R2 stages (Liu et al., 2011). However, as we show in Eq. (1) and Sect. 2.1, crops in LPJ-GUESS are described by crop functional types (CFTs), only with two main growing stages being modelled (i.e., vegetative and reproductive phases). Thus varying C cost per unit fixed N between V4-R2 stages unfortunately cannot be represented in the model.

*Regarding crop phenology days, authors used field-measured phu (Eq.1, it's actually 'degree days' if I understand correctly) to represent soybean growing period, Is the phu a fixed or dynamic value across all evaluated sites? I'm just aware that the discrepancy in various phu values basically has huge impacts on harvested yield and accumulative N fixation because of the different growing periods. Also, how does LPJ-G represent the crop growing period on global scale? Sowing date and harvest date? Or multi-cropping systems in tropics are considered when comparing with FAO statistics?*

Similar to most ecosystem and crop models, LPJ-GUESS adopts crop-specific accumulated heat requirements to model plant growth development, and crops are allowed to adapt to the local climate by dynamically adjusting by the heat requirements (potential heat units, phu) to different climatic zones (Lindeskog et al., 2013).

Regarding the representation of the crop growing period on regional scale, LPJ-GUESS adopts the dynamic sowing dates based on local climate with five seasonality types incorporated (Waha et al., 2012). The five seasonality types are determined by temperature and precipitation conditions, with the intra-annual variability of temperature and precipitation being especially important. We applied specific rules per seasonality type to simulate sowing date. For example, in the region with precipitation seasonality, we assumed that crops sow at the onset of the main rainy season, which is defined as the largest sum of monthly precipitation-to-potential-evapotranspiration ratios of four consecutive months (Lindeskog et al., 2013). In the model croplands are harvested each year when prescribed heat sum requirements are fulfilled. Multi-cropping systems within a year are not yet implemented in LPJ-GUESS.

We added the information on the calculation of sowing and harvest dates to Sect. 2.1 and 2.2 in the new manuscript: "Sowing dates on a large scale are determined dynamically in the model based on local climatology in each grid cell with five seasonality types represented (combination of temeperature and precipiation limited behaviors, Waha et al., 2012), and crops are harvested once each year when prescribed heat sum requirements are fulfilled (Lindeskog et al., 2013). Multi-cropping systems within a year are not yet implemented in the model."

*Authors highlighted that their model performed better in the top10 soybean-producing countries, but what about other countries? Look at the map of yield bias between simulation and observation (Fig. 9), the reported yields in most African countries were almost overestimated by 300%, any explanations? Also in terms of N fixation map in Africa, the simulated BNF rate can be as high as 300 kg N/ha (Fig. 10), this value is far from reality and does not make sense to me. The N fixation in African smallholder is greatly variable, even fixing zero N from the atmosphere sometimes because of the acid soil (Vanlauwe et al., 2019). Moreover, using grain legumes as green manure to replace the industrial N-fertilizer is not common in African and east Asian farmer's fields (N fixing grasses are much popular), It'll be much interesting to add relevant discussion on modelling forage legumes.*

In our simulations, legume yields are strongly and positively correlated to the rate of N fixation across a range of field sites, agreeing well with the observations (see Sect. 3.1.1). Regionally, LPJ-GUESS mainly overestimated yield production in some African countries, due to the high simulated BNF rates in these regions (see Fig.10). Similar to the reviewer perspective, more recent studies from African farms found that the soybean N fixation rate can be very low, as a result of the inconsistent effectiveness of inoculation in the acid soils (Ulzen et al., 2016; Vanlauwe et al., 2019). Unfortunately, the BNF implemented in the model do not account for inoculation effectiveness in response to soil pH at the moment, due to inadequate amount of information available to establish a reliable relationship between BNF and this important limitation factor.

To clarify the reasons, we gave the relevant explanations and discussions in Sect. 4.2: "LPJ-GUESS produced an overestimation of 100-300% in yield production among some African countries, with BNF rates of 300-350 kg N ha$^{-1}$ yr$^{-1}$ being

modelled in these regions (Fig. 10a). More recent studies that report data from African farms have indicated that the soybean N fixation rate can be as low as 0-50 kg N ha$^{-1}$ yr$^{-1}$ in most farmers' fields, largely because of the inconsistent effectiveness of inoculation in the acid soils (Ulzen et al., 2016; Muleta et al., 2017; Vanlauwe et al., 2019). The BNF implementation and soil representation in LPJ-GUESS do not account for inoculation effectiveness in response to soil pH."

We do agree with the reviewer's opinion that using herbaceous legumes as cover crops is much more popular in African and East Asian cropping systems, compared to the grain legumes. The evaluation and regional simulation of N-fixing grasses effects on the cropland C-N cycle will be presented and discussed in a forthcoming paper separately; this study only concentrates on discussing N fixation in global grain legumes.

**Specific comments:**

*Line 50-65: is it necessary to list the model name in the brackets?*

Model names in brackets will be removed as suggested.

*Line 94: "two temperate C3 crops with sowing carried out in spring and autumn", what is the specific name of these two C3 crops? wheat?*

Crops in LPJ-GUESS are modelled as crop functional types (CFTs), i.e. group of crops with similar functional behaviors. Temperate C3 crops in the model generally stands for wheat, rye, barley, etc.

*Line 100: "faba bean as a second as a representation of pulses more generally", it's unclear to me, you mean to simulate other pulses using faba bean parameters?*

See previous comment. The faba bean parametrization in this study is used to more generally represent all the pulses.
Revision in sentence: "the parametrization of faba bean is representative for the group of pulses in general"

*Line 150-152 (Fig. 1): Interesting findings, is the data from Penning De Vries et al. (1989) the only source for fitting the soybean assimilate partitioning in the study? Any other possible measurement for comparison?*

Yes, data from Penning de Vries et al. (1989) is the only source for fitting the soybean assimilate partitioning. We haven't found any other potential sources to compare with Penning De Vries et al. (1989).

*Line 185: why is soil temperature at 25 cm depth? No nitrogenase activity below the 25cm of depth?*

LPJ-GUESS uses soil temperature at 25cm to represent the top layer of 0-50cm. Root nodulation in grain legumes is mainly observed at plough layer (0-30cm) in most field experiments (Juge et al., 2012). We thus assumed that nitrogenase activity in the model only takes place in the topsoil (0-50cm).
The sentence is revised as "where $T$ is soil temperature (°C) at 25 cm depth, representing the mean temperature of the topsoil layer (0-50cm) in the model."

*Line 213-215 (Eq.13): To give readers a general idea on how MAXbnfcost is varying over the growing season, would be better to plot its function somewhere. Maybe in Fig. S5 in the Supplement.*

Thank you for this suggestion. We added the MAX$_{bnfcost}$ below to the revised Fig. S7b.

[Figure]

**Figure S7.** Modelled BNF-limiting factors (a) and daily NPP for N fixation cost (b) in nodulating soybean treatment in the 1984 cropping season at an Austrian site (Zapata et al., 1987). Veg. and Rep. indicate vegetative and reproductive growth phase, respectively. The reported GPP cost range of N fixation (4-16%) was extracted from Table 1 given in Kaschuk et al. (2009), and converted to NPP cost (8-32%) by multiplying by 2.0, assuming GPP=2×NPP (Waring et al., 1998).

*Line 245: 'to convert plant C mass to dry matter, a conversion factor of 2.0 was used'. The conversion factor of 2.0 seems a little bit lower than that data based on the tons of published literature. did it impact the conclusion of the paper?*

Compared to the published measurement of 2.24 reported in Osaki (1993), the conversion factor of 2.0 used in the model could underestimate the observed yields by 10%. This can partially explain that the productivity generally is somewhat low in the model (see Fig. 4a-b). However, this conversion factor will not affect the main conclusions of our paper.

We gave the relevant discussion in Sect. 4.1: "A similar, small, underestimation was found in the shoot N mass (Fig. 4c), indicating that the productivity generally is somewhat too low in the model. One factor contributing to the underestimation is that LPJ-GUESS applies a conversion factor of 2.0 from plant C mass to dry matter (Smith et al., 2014), ~10% lower than a published measurement of 2.24 reported in Osaki (1993)"

*Line 255-257: 'Gridded daily climate data ......used from GSWP3-W5E5', why not field-based weather data? I understand the observed climate may not available at some sites, but the discrepancy in climate may bias the evaluation results. I suggest authors make a comparison between gridded climate and observed data at several field sites, where the recorded weather data are available.*

There are two reasons to use the GSWP3-W5E5 data set instead of field-based weather records: (1) As the reviewer mentions, information on weather data is not available at the majority of the sites evaluated; (2) the historical soil C-N pools prior to the field experiments are important for modelling crop growth. For evaluation, we performed the simulations from 1901 to the year before the field trials start, with long-term historical climate as inputs, with such long time series not being available from field-based weather data. GSWP3-W5E5 data, as an alternative of field-based climate, can meet all the requirement for our simulations. We did compare model-required input variables (e.g., daily mean temperature and solar radiation) from GSWP3-W5E5 with observations at three sites (see Figure below), finding that the gridded climate data had a fairly good agreement with weather records in the field, despite some radiation deviations between two data sets for individual days over the experimental period. This figure is added to the SI of the revised manuscript.

[Figure]

**Figure S2.** Comparison of daily climate between GSWP3-W5E5 data set and field-based weather records over the experimental period at USA-Illinois (a), Spain-Lugo (b), and Kenya- Kisumu sites (c). RB and AB are mean relative bias (Eq. 16) and absolute bias (Eq. 17), respectively, presented in percent (%). See Table S2 for the BNF trials of these three sites.

*Line 285: would be helpful to plot the map of N fertilizer (mineral + manure) applied to soybean and add it to the Supplement.*

We thank the reviewer for this suggestion and have added the desired figure to the SI of the revised version of the manuscript. The following figure shows the global map of mineral N fertilizer (a) and total fertilizer inputs (mineral N + manure) (b) for soybean and pulses, averaged over 1996-2005 (kg N ha$^{-1}$).

[Figure]

**Figure S4.** Global map of mineral N fertilizer (a) and total fertilizer inputs (mineral N + manure) (b) for soybean (top) and pulses (bottom), averaged over 1996-2005 (kg N ha$^{-1}$, see Sect. 2.4.2).

*Line 345: I noticed that soybean shoot biomass was largely underestimated in Fig.5a, is it a common result across most of sites or only happening at the U.S. site? If it is common among all evaluated sites, authors should check their fitted assimilate partitioning in Fig.1 furthermore, and the C allocation scheme to plant organs should be further improved.*

In our simulations, underestimated shoot biomass is found in three sites. One possible explanation is that LPJ-GUESS at this point does not model soybean hulls, which account for ~15-20% of the total above-ground dry matter at harvest in the soybean rain-fed cropping system (Córdova et al., 2020). We agree with the reviewer's opinion that the C allocation scheme plays an important role in modelling crop production, in particular for the root vs. shoot allocation (i.e., Eq. 5). Unfortunately, an evaluation is mainly prevented by the insufficient information on observed root biomass in this study.

We gave the relevant discussion in Sect. 4.1: "In addition, we found that the model underestimated above-ground biomass while simultaneously overestimating below-ground productivity at the three sites where measured root biomass was available. This could be addressed by adjusting the root vs. shoot allocation (i.e., modifying the daily assimilate partitioning function in grain legumes; Eq. 5), but this is currently prevented by the lack of sufficient observed root biomass information."

*Table 2: For the observed and modelled soil N uptake, why is non-nodulation experiment significantly higher than nodulation one? different N-fertilizer application?*

Yes, the difference can be attributed to N fertilizer application. According to Zapata et al. (1987), application rates of 33 and 100 kg N ha$^{-1}$ were applied to nodulation and non-nodulation treatments, respectively.

*Line 397: 'as a consequence of the inoculation implemented.....', just want make it clear to me: inoculation is only available in the unfertilized treatment, but not in the N300 treatment; thus N fixation from N0 is greater than N300, resulting in slight difference in total N uptake between two treatments?*

Correct.

*Line 417: 'Modelled low yields were found in some arid and semi-arid countries (e.g., Egypt, Iran, and Turkey)', the reason is water constraint on photosynthesis or on BNF or both? Look at the map of environmental limitation to BNF (Fig. S4, in the Supplement), in these regions water is a key factor affecting N fixation.*

Water stress in the model affects photosynthesis by constraining $V_{max}$ (the maximum carboxylation activity of Rubisco, Smith et al., 2014; Olin et al., 2015), and it also limits N fixation by inhibiting nitrogenase activity (see Eq.11). Both of these factors in LPJ-GUESS contribute to low crop yields in the arid and semi-arid regions.

*Line 485: As I mentioned earlier, the conversion factor of 2.0 is low, authors also discussed here. Why not use the new factor of 2.24 in the model evaluation?*

The factor of 2.0 is used not only to convert yield from C mass to dry matter, but also to calculate N fixation via root biomass (see Eq. 9). To keep the consistency of biomass computation in the model, the C-N version of LPJ-GUESS in this study adopts the uniform value of 2.0. The new parameter (i.e., 2.24) would be most likely used in the next model version. However, as stated above, the conversion factor of 2.0 currently implemented in the model will not affect the main conclusions in our paper.

*Line 542: 'with the reported range of 14-32%', why the cost fraction becomes to 14-32% here? it is described as 4-16% in the Instruction section (see Line 59)*

See previous comment and changes implemented to make this clear.

*Line 557: 'with the estimate of 19.4 Tg provided by Herridge et al. (2008)' . I don't think 19.4 Tg N is the correct value derived from Herridge et al. (2008), in which a total of 21.45 Tg N is fixed by all legume crops every year globally.*

Thanks for this comment. Herridge et al. (2008) estimated a total annual N fixation of 21.45 Tg N across all legume crops globally, with 2.95, 16.44 and 2.06 Tg N fixed by pulses, soybean and groundnut, respectively. However, our study does not include groundnut, which belongs to oilseeds, rather than grain legumes. Thus, we use the estimate of 19.4 Tg N (pulses + soybean) in Herridge et al. (2008) for comparison in our study.

**References**

Ciampitti, I. A., de Borja Reis, A. F., Córdova, S. C., Castellano, M. J., Archontoulis, S. V., Correndo, A. A., Antunes De Almeida, L. F. and Moro Rosso, L. H.: Revisiting Biological Nitrogen Fixation Dynamics in Soybeans, Front. Plant Sci., 12(October), 1–11, doi:10.3389/fpls.2021.727021, 2021.

Córdova, S. C., Archontoulis, S. V. and Licht, M. A.: Soybean profitability and yield component response to nitrogen fertilizer in Iowa, Agrosystems, Geosci. Environ., 3(1), 1–16, doi:10.1002/agg2.20092, 2020.

Elliott, J., Müller, C., Deryng, D., Chryssanthacopoulos, J., Boote, K. J., Büchner, M., Foster, I., Glotter, M., Heinke, J., Iizumi, T., Izaurralde, R. C., Mueller, N. D., Ray, D. K., Rosenzweig, C., Ruane, A. C. and Sheffield, J.: The Global Gridded Crop Model Intercomparison: Data and modeling protocols for Phase 1 (v1.0), Geosci. Model Dev., 8(2), 261–277, doi:10.5194/gmd-8-261-2015, 2015.

Herridge, D. F., Peoples, M. B. and Boddey, R. M.: Global inputs of biological nitrogen fixation in agricultural systems, Plant Soil, 311(1–2), 1–18, doi:10.1007/s11104-008-9668-3, 2008.

Jiang, S., Jardinaud, M., Gao, J., Pecrix, Y., Wen, J., Mysore, K., Xu, P., Sanchez-canizares, C., Ruan, Y., Li, Q., Zhu, M., Li, F., Wang, E., Poole, P. S., Gamas, P. and Murray, J. D.: NIN-like protein transcription factors regulate leghemoglobin genes in legume nodules, Science (80-. )., 374, 625–628, 2021.

Juge, C., Prévost, D., Bertrand, A., Bipfubusa, M. and Chalifour, F. P.: Growth and biochemical responses of soybean to double and triple microbial associations with Bradyrhizobium, Azospirillum and arbuscular mycorrhizae, Appl. Soil Ecol., 61, 147–157, doi:10.1016/j.apsoil.2012.05.006, 2012.

Kaschuk, G., Kuyper, T. W., Leffelaar, P. A., Hungria, M. and Giller, K. E.: Are the rates of photosynthesis stimulated by the carbon sink strength of rhizobial and arbuscular mycorrhizal symbioses?, Soil Biol. Biochem., 41(6), 1233–1244, doi:10.1016/j.soilbio.2009.03.005, 2009.

Kaschuk, G., Hungria, M., Leffelaar, P. A., Giller, K. E. and Kuyper, T. W.: Differences in photosynthetic behaviour and leaf senescence of soybean (Glycine max [L.] Merrill) dependent on N2 fixation or nitrate supply, Plant Biol., 12(1), 60–69, doi:10.1111/j.1438-8677.2009.00211.x, 2010.

Lindeskog, M., Arneth, A., Bondeau, A., Waha, K., Seaquist, J., Olin, S. and Smith, B.: Implications of accounting for land use in simulations of ecosystem carbon cycling in Africa, Earth Syst. Dyn., 4(2), 385–407, doi:10.5194/esd-4-385-2013, 2013.

Liu, Y., Wu, L., Baddeley, J. A. and Watson, C. A.: Models of biological nitrogen fixation of legumes. A review, Agron. Sustain. Dev., 31(1), 155–172, doi:10.1051/agro/2010008, 2011.

Marino, D., Frendo, P., Ladrera, R., Zabalza, A., Puppo, A., Arrese-Igor, C. and Gonzalez, E. M.: Nitrogen Fixation Control under Drought Stress. Localized or Systemic?, Plant Physiol., 143(4), 1968–1974, doi:10.1104/pp.106.097139, 2007.

Mourtzinis, S., Kaur, G., Orlowski, J. M., Shapiro, C. A., Lee, C. D., Wortmann, C., Holshouser, D., Nafziger, E. D., Kandel, H., Niekamp, J., Ross, W. J., Lofton, J., Vonk, J., Roozeboom, K. L., Thelen, K. D., Lindsey, L. E., Staton, M., Naeve, S. L., Casteel, S. N., Wiebold, W. J. and Conley, S. P.: Soybean response to nitrogen application across the United States: A synthesis-analysis, F. Crop. Res., 215, 74–82, doi:10.1016/j.fcr.2017.09.035, 2018.

Olin, S., Schurgers, G., Lindeskog, M., Wärlind, D., Smith, B., Bodin, P., Holmér, J. and Arneth, A.: Modelling the response of yields and tissue C : N to changes in atmospheric CO2 and N management in the main wheat regions of western Europe, Biogeosciences, 12(8), 2489–2515, doi:10.5194/bg-12-2489-2015, 2015.

Osaki, M.: Carbon-nitrogen interaction model in field crop production, Plant Soil, 155–156(1), 203–206, doi:10.1007/BF00025019, 1993.

Penning de Vries, F. W. T., Jansen, D. M., Berge, H. F. M. ten and Bakema, A.: Simulation of ecophysiological processes of growth in several annual crops, Centre for Agricultural Publishing and Documentation, Wageningen, Wageningen., 1989.

Santachiara, G., Borrás, L. and Rotundo, J. L.: Physiological processes leading to similar yield in contrasting soybean maturity groups, Agron. J., 109(1), 158–167, doi:10.2134/agronj2016.04.0198, 2017.

Serraj, R., Sinclair, T. R. and Purcell, L. C.: Symbiotic N2 fixation response to drought, J. Exp. Bot., 50(331), 143–155, doi:10.1093/jxb/50.331.143, 1999.

Smith, B., Wärlind, D., Arneth, A., Hickler, T., Leadley, P., Siltberg, J. and Zaehle, S.: Implications of incorporating N cycling and N limitations on primary production in an individual-based dynamic vegetation model, Biogeosciences, 11(7), 2027–2054, doi:10.5194/bg-11-2027-2014, 2014.

Ulzen, J., Abaidoo, R. C., Mensah, N. E., Masso, C. and AbdelGadir, A. A. H.: Bradyrhizobium inoculants enhance grain yields of soybean and cowpea in Northern Ghana, Front. Plant Sci., 7, 1–9, doi:10.3389/fpls.2016.01770, 2016.

Vanlauwe, B., Hungria, M., Kanampiu, F. and Giller, K. E.: The role of legumes in the sustainable intensification of African smallholder agriculture: Lessons learnt and challenges for the future, Agric. Ecosyst. Environ., 284(December), 106583, doi:10.1016/j.agee.2019.106583, 2019.

Waha, K., Van Bussel, L. G. J., Müller, C. and Bondeau, A.: Climate-driven simulation of global crop sowing dates, Glob. Ecol. Biogeogr., 21(2), 247–259, doi:10.1111/j.1466-8238.2011.00678.x, 2012.

Waring, R. H., Landsberg, J. J. and Williams, M.: Net primary production of forests: A constant fraction of gross primary production?, Tree Physiol., 18(2), 129–134, doi:10.1093/treephys/18.2.129, 1998.

Wu, L. and McGechan, M. B.: Simulation of nitrogen uptake, fixation and leaching in a grass/white clover mixture, Grass Forage Sci., 54(1), 30–41, doi:10.1046/j.1365-2494.1999.00145.x, 1999.

Zapata, F., Danso, S. K. A., Hardarson, G. and Fried, M.: Time Course of Nitrogen Fixation in Field-Grown Soybean Using Nitrogen-15 Methodology, Agron. J., 79(1), 172–176, doi:10.2134/agronj1987.00021962007900010035x, 1987.

Zhang, B., Tian, H., Lu, C., Dangal, S., Yang, J. and Pan, S.: Manure nitrogen production and application in cropland and rangeland during 1860–2014: A 5-minute gridded global data set for Earth system modeling, Earth Syst. Sci. Data, 9, 667–678, doi:10.5194/essd-2017-11, 2017.

**Response to anonymous referee #2** (Comments in black, Answers in blue, Revisions in red)

**General comments:**

*Ma et al. describe a new parameterisation for two types of nitrogen fixing crops in tropical and temperate regions for the widely used LPJ-Guess model. They clearly explain and document the added mathematical formulation and parameters, provide a succinct evaluation of the model using suitable data and also provide a global evaluation of the consequences of the model implementation in terms of simulated nitrogen fixation and yields. Overall this is a well written paper, which only requires minor adjustments. The model description is sufficient to understand what has been done and tested, but I want to highlight that the manuscript falls short of making the code accessible to the public. The code availability statement says that the code would be made available for review, but I have not been provided with such a link.*

We thank the reviewer for providing constructive feedback and apologize for the inaccessible code during the review process due to the link having expired in mid-October. The new code link via SVN access in Lund university is svn://stormbringer.nateko.lu.se/svn/LPJ-GUESS/tags/_publications/newcrops_bnf_evaluation-gmd-2021-260. In the revision to the manuscript, we address all of the points raised by the reviewer, and we think the resulting study is much improved. Specific revisions and responses to each comment are provided in detail below.

**Minor comments:**

*A few model choices have been made that deviate from previous approaches (e.g. how N fix responds to growth stages, that the cap of NPP investment to N fix is, why is oxygen required for N fixation, which is a process happening in anoxic environments, and why is oxygen availability ignored in the water limitation function). It would be helpful for readers and fellow model developers to better understand the motivation of these choices. In each case, probably one - two sentence explaining the choice would be sufficient.*

We thank the reviewer for this suggestion. Brief explanations on main BNF-limiting factors is added to the revised manuscript, as follows.

For BNF response to growth stage (Sect. 2.3): "Much experimental evidence has indicated that the N fixed by legumes varies widely among crop growth stages, with the largest BNF rate observed between the late vegetative phase and the early seed-filling period (Santachiara et al., 2017; Córdova et al. 2020; Ciampitti et al., 2021). In this study, a specific function, similar to the temperature response function, is thus implemented in the BNF scheme to represent the variation of N fixation with the course of legume life cycle."

For water limitation (Sect. 2.3): "Too little water strongly inhibits BNF due to impacts of drought stress on nodule nitrogenase activity (Serraj et al., 1999; Marino et al., 2007). Although oxygen is needed to support the respiration of legume roots and bacteria in the nodules, nitrogenase is more active in anoxic, waterlogged environments (Jiang et al., 2021). A linear water-limitation function is thus incorporated into LPJ-GUESS (Wu and McGechan, 1999)."

As the reviewer mentions, the implementation of the NPP cost for BNF in our study is different from previous studies. Thus, we would like to give an in-depth discussion in Sect. 4.1: "N fixation can help grain legumes to dramatically enhance their total N accumulation and to achieve higher N concentration in seeds. However, these benefits are accompanied by an increase in respiration cost amounting to 4-16% of fixed total photosynthetic carbon (Kaschuk et al., 2009, 2010). Such a respiratory photosynthate consumption would reduce productivity if photosynthesis rate was not increased to compensate for the cost. In LPJ-GUESS, as described in Sect. 2.3, we assumed that up to 50% of daily NPP can be consumed to fix N. This approach has the advantage that legumes are able to maximize photosynthetic gain due to reduced N limitation in carboxylation capacity ($V_{max}$), but it entails the risk of lower productivity if too much NPP is invested into fixation. Nevertheless, in most cases our modelled NPP cost over the soybean growing season was ranging from 1-40 % at site scale (Fig. S7) and 5-25% on a large region (Fig. S8). Such NPP consumption was not only lower than our assumed upper limit of 50%, but also appropriately consistent with the reported

range of 14-32% described by Kaschuk et al. (2009), demonstrating that the C cost scheme implemented for N fixation in our model is reasonable."

*A somewhat more comprehensive explanation on the calculation of planting dates would be appreciated.*

We agree with the reviewer's opinion that a comprehensive description on the representation of sowing dates would be helpful for readers to understand the crop growth in the model. Generally, LPJ-GUESS adopts the dynamic sowing dates based on local climate with five seasonality types incorporated (Waha et al., 2012). The five seasonality types are determined by temperature and precipitation conditions, with the intra-annual variability of temperature and precipitation being especially important. We applied specific rules per seasonality type to simulate sowing date. For example, in the region with temperature seasonality, sowing starts when daily average temperature exceeds a prescribed crop-specific threshold. In the region with precipitation seasonality, we assumed that crops sow at the onset of the main rainy season, which in the model is defined as the largest sum of monthly precipitation-to-potential-evapotranspiration ratios of four consecutive months (Lindeskog et al., 2013). More details can refer to the description in Waha et al. (2012).

We added a brief explanation of crop growth periods in Sect. 2.1 in the new manuscript. Suggested revision: "Sowing dates on a large scale are determined dynamically in the model based on local climatology in each grid cell with five seasonality types represented (combination of temeperature and precipiation limited behaviors, Waha et al., 2012), and crops are harvested once each year when prescribed heat sum requirements are fulfilled (Lindeskog et al., 2013). "

*The method description is not explicit about the source of N fertiliser used. This information should be added to Section 2.4. It is specifically important to clarify how the authors have dealt with the N fertiliser information that is related to cropland N fixation, which is included as a factor in many estimates of N fertiliser application.*

Globally, time-dependent mineral N fertilizer and manure used in this study are taken from Ag-GRID (AgMIP GRIDded Crop Modeling Initiative; Elliott et al. (2015) and Zhang et al. (2017), respectively). The application rates of these two N fertilizer types to soybean and pulses are given in the figure below, averaged over the period of 1996-2005 (kg N ha$^{-1}$). The figure is added to the SI of the revised manuscript.

[Figure]

**Figure S4.** Global map of mineral N fertilizer (a) and total fertilizer inputs (mineral N + manure) (b) for soybean (top) and pulses (bottom), averaged over 1996-2005 (kg N ha$^{-1}$, see Sect. 2.4.2).

We also added the N fertilizer information in Sect. 2.4.2 as suggested, including the timing and amount of application, as well as the fertilizer sources.Revision: "In terms of timing of N fertilizer application, a recent meta-analysis conducted by Mourtzinis et al. (2018) indicated that splitting N application between planting and the early reproductive stage resulted in significantly greater soybean yields than a single application. Mineral N fertilizer for legumes was thus split into two equal applications at the time of sowing (DS=0) and flowering (DS=1.0). Manure was added to soils at the time of sowing as a single application to reflect real-world practices that account for the time required for manure N to be made available to plants. Data sources for mineral N fertilizer and manure over the period 1901-2014 were derived from Ag-GRID (AgMIP GRIDded Crop Modeling Initiative; Elliott et al. (2015) and Zhang et al. (2017), respectively) (Fig. S4)."

*Figure 4: Please be specific, which regression data belongs to which simulation set.*

To make it clearer to readers, we updated the legend of Figure 4 and add the relevant explanations on linear regression in caption.

Revision for the caption of figure 4: "Comparison of modelled and observed yield (a), grain N mass (b), shoot N mass (c), soil N uptake (d), BNF (e) and %Ndfa (the proportion of plant N derived from the atmosphere) (f) at harvest across all soybean and faba bean sites. Filled red and grey circles depict the 'site-specific' and 'global-uniform' runs, respectively. The dashed line is fitted linear regression with red for 'site-specific' and grey for 'global-uniform'; *** and ** denote regressions statistically significant at p=0.001 and 0.01, respectively; AB is absolute bias (Eq. 17), represented in percent (%); the unit of RMSE is the same as the associated variable; AVG in (f) is the averaged value of %Ndfa across all field trials".

*Figure 6: does panel e not suggest that there's something wrong in the time dependence of N fixation?*

In the real world, it might take a while after inoculation for nodulation to happen and fixation to reach full capacity, but in LPJ-GUESS we assume that all the fixation machinery is there and ready to work after a few days of planting (see Eq.12). Another possible explanation is the difference between the simulated and observed crop growing stage at this Austrian soybean site, mainly due to the use of GSWP3-W5E5 climate data set in our simulation (see Sect. 2.4.1, because of the unavailability of field-based weather records at majority of sites evaluated). The daily mean temperature from this reanalysis data set is most likely differing from the field observations because the grid cell at 0.5° coarse resolution in GSWP3-W5E5 may not realistically represent the field-based weather conditions on the fine scale (see Figure below, we compared the climate difference between GSWP3-W5E5 and field-based records at three sites). The temperature difference would affect the legume development stage via accumulated heat units (see Eq.1), and subsequently bias the daily pattern of N fixation in the model. However, from the perspective of the entire growing season, our modelled total N fixation is close to the measured value at harvest (see Table 2).

[Figure]

**Figure S2.** Comparison of daily climate between GSWP3-W5E5 data set and field-based weather records over the experimental period at USA-Illinois (a), Spain-Lugo (b), and Kenya- Kisumu sites (c). RB and AB are mean relative bias (Eq. 16) and absolute bias (Eq. 17), respectively, presented in percent (%). See Table S2 for the BNF trials of these three sites.

*Figure 9: Please attempt to arrange the panels more logically (e.g. placement of China and Canada?).*

We arranged the panels of top 10 soybean-producing countries (i.e., b-k) based on their total production from 1996-2005. For example, the largest production country (U.S.A.) is labeled as (b), the second largest producer (Brazil) is labeled as (c), etc. To make it clearer to readers, we add the explanations in the figure caption.

Revision: "Comparison of simulated and FAO-reported yields on country level averaged over 1996–2005 (a), as well as time series of modelled soybean yield (red solid line) and reported FAO statistics (black dashed line) in the top 10 producer countries over the period 1981–2016. The top 10 producer countries (b–k, in descending order) were chosen based on their total production from 1996–2005. $r$ is Pearson correlation coefficient (Eq. 19), where ***, ** and * denote the correlation to be statistically significant at p=0.001, 0.01, and 0.05 level, respectively. RB is relative bias (Eq. 16), represented in percent (%). $SD_{Rep}$ and $SD_{Mod}$ denote, respectively, reported and modelled yield standard deviations (t ha$^{-1}$ yr$^{-1}$) from 1981-2016".

*Figure 10: Here it is important to understand what the underlying N fertilisation data source was, and to which extend this contributes to finding?*

As explained above, a global map of N fertilizer inputs is added to the SI of the revised manuscript, and data source information will be added to Sect. 2.4.2 as suggested, including the timing and amount of application. We agree with the reviewer opinion that

N fertilizer could affect the modelled N fixation rate regionally. For instance, the low simulated soybean %Ndfa (the proportion of plant N derived from the atmosphere) in East Asia may reflect high N uptake from soils in response to substantial fertilizer investment in China (80-180 kg N ha$^{-1}$ yr$^{-1}$, see the N fertilizer map above). In contrast, the high modelled %Ndfa in Africa was found in our simulations. We will revise the relevant sentences in the main text to highlight the contribution of N fertilizer to our findings.

Related revision in Sect. 3.2.2: "Moreover, the relatively low fertilizer application in Africa (0-20 kg N ha$^{-1}$ yr$^{-1}$, Fig. S4b) leaves a nitrogen deficit that causes enhanced soybean N fixation. In contrast, in arid and semi-arid regions, soil water constrains BNF, while temperature limitation is seen in high latitudes and alpine areas (e.g., Andes in Peru). BNF rates in most regions (South Asia, West Asia, Sub-Saharan Africa and northwest China) were as low as 50 kg N ha$^{-1}$ yr$^{-1}$, particularly in Pakistan and northern India, where simulated BNF is severely constrained by the extreme high temperature over the cropping season. Eastern United States, Europe, Southern China and central-west Brazil showed intermediate fixation rates, which were greater than 150 kg N ha$^{-1}$ yr$^{-1}$. Overall, the spatial variation of modelled legume BNF rate reflects to large degree the spatial climate patterns, in addition to N-fertilizer application. The low modelled %Ndfa of 45±3% in East Asia may reflect high N uptake from soils in response to substantial fertilizer investment in China (80-180 kg N ha$^{-1}$ yr$^{-1}$, Fig. S4b) over the past 40 years. In contrast, the modelled %Ndfa in Africa—with lower N application rates—was as high as 70±3%, although still lower than the reported mean value of 77% (Table 3)".

**References**

Ciampitti, I. A., de Borja Reis, A. F., Córdova, S. C., Castellano, M. J., Archontoulis, S. V., Correndo, A. A., Antunes De Almeida, L. F. and Moro Rosso, L. H.: Revisiting Biological Nitrogen Fixation Dynamics in Soybeans, Front. Plant Sci., 12(October), 1–11, doi:10.3389/fpls.2021.727021, 2021.

Córdova, S. C., Archontoulis, S. V. and Licht, M. A.: Soybean profitability and yield component response to nitrogen fertilizer in Iowa, Agrosystems, Geosci. Environ., 3(1), 1–16, doi:10.1002/agg2.20092, 2020.

Elliott, J., Müller, C., Deryng, D., Chryssanthacopoulos, J., Boote, K. J., Büchner, M., Foster, I., Glotter, M., Heinke, J., Iizumi, T., Izaurralde, R. C., Mueller, N. D., Ray, D. K., Rosenzweig, C., Ruane, A. C. and Sheffield, J.: The Global Gridded Crop Model Intercomparison: Data and modeling protocols for Phase 1 (v1.0), Geosci. Model Dev., 8(2), 261–277, doi:10.5194/gmd-8-261-2015, 2015.

Herridge, D. F., Peoples, M. B. and Boddey, R. M.: Global inputs of biological nitrogen fixation in agricultural systems, Plant Soil, 311(1–2), 1–18, doi:10.1007/s11104-008-9668-3, 2008.

Jiang, S., Jardinaud, M., Gao, J., Pecrix, Y., Wen, J., Mysore, K., Xu, P., Sanchez-canizares, C., Ruan, Y., Li, Q., Zhu, M., Li, F., Wang, E., Poole, P. S., Gamas, P. and Murray, J. D.: NIN-like protein transcription factors regulate leghemoglobin genes in legume nodules, Science (80-. )., 374, 625–628, 2021.

Juge, C., Prévost, D., Bertrand, A., Bipfubusa, M. and Chalifour, F. P.: Growth and biochemical responses of soybean to double and triple microbial associations with Bradyrhizobium, Azospirillum and arbuscular mycorrhizae, Appl. Soil Ecol., 61, 147–157, doi:10.1016/j.apsoil.2012.05.006, 2012.

Kaschuk, G., Kuyper, T. W., Leffelaar, P. A., Hungria, M. and Giller, K. E.: Are the rates of photosynthesis stimulated by the carbon sink strength of rhizobial and arbuscular mycorrhizal symbioses?, Soil Biol. Biochem., 41(6), 1233–1244, doi:10.1016/j.soilbio.2009.03.005, 2009.

Kaschuk, G., Hungria, M., Leffelaar, P. A., Giller, K. E. and Kuyper, T. W.: Differences in photosynthetic behaviour and leaf senescence of soybean (Glycine max [L.] Merrill) dependent on N2 fixation or nitrate supply, Plant Biol., 12(1), 60–69, doi:10.1111/j.1438-8677.2009.00211.x, 2010.

Lindeskog, M., Arneth, A., Bondeau, A., Waha, K., Seaquist, J., Olin, S. and Smith, B.: Implications of accounting for land use in

simulations of ecosystem carbon cycling in Africa, Earth Syst. Dyn., 4(2), 385–407, doi:10.5194/esd-4-385-2013, 2013.

Liu, Y., Wu, L., Baddeley, J. A. and Watson, C. A.: Models of biological nitrogen fixation of legumes. A review, Agron. Sustain. Dev., 31(1), 155–172, doi:10.1051/agro/2010008, 2011.

Marino, D., Frendo, P., Ladrera, R., Zabalza, A., Puppo, A., Arrese-Igor, C. and Gonzalez, E. M.: Nitrogen Fixation Control under Drought Stress. Localized or Systemic?, Plant Physiol., 143(4), 1968–1974, doi:10.1104/pp.106.097139, 2007.

Mourtzinis, S., Kaur, G., Orlowski, J. M., Shapiro, C. A., Lee, C. D., Wortmann, C., Holshouser, D., Nafziger, E. D., Kandel, H., Niekamp, J., Ross, W. J., Lofton, J., Vonk, J., Roozeboom, K. L., Thelen, K. D., Lindsey, L. E., Staton, M., Naeve, S. L., Casteel, S. N., Wiebold, W. J. and Conley, S. P.: Soybean response to nitrogen application across the United States: A synthesis-analysis, F. Crop. Res., 215, 74–82, doi:10.1016/j.fcr.2017.09.035, 2018.

Olin, S., Schurgers, G., Lindeskog, M., Wärlind, D., Smith, B., Bodin, P., Holmér, J. and Arneth, A.: Modelling the response of yields and tissue C : N to changes in atmospheric CO2 and N management in the main wheat regions of western Europe, Biogeosciences, 12(8), 2489–2515, doi:10.5194/bg-12-2489-2015, 2015.

Osaki, M.: Carbon-nitrogen interaction model in field crop production, Plant Soil, 155–156(1), 203–206, doi:10.1007/BF00025019, 1993.

Penning de Vries, F. W. T., Jansen, D. M., Berge, H. F. M. ten and Bakema, A.: Simulation of ecophysiological processes of growth in several annual crops, Centre for Agricultural Publishing and Documentation, Wageningen, Wageningen., 1989.

Santachiara, G., Borrás, L. and Rotundo, J. L.: Physiological processes leading to similar yield in contrasting soybean maturity groups, Agron. J., 109(1), 158–167, doi:10.2134/agronj2016.04.0198, 2017.

Serraj, R., Sinclair, T. R. and Purcell, L. C.: Symbiotic N2 fixation response to drought, J. Exp. Bot., 50(331), 143–155, doi:10.1093/jxb/50.331.143, 1999.

Smith, B., Wärlind, D., Arneth, A., Hickler, T., Leadley, P., Siltberg, J. and Zaehle, S.: Implications of incorporating N cycling and N limitations on primary production in an individual-based dynamic vegetation model, Biogeosciences, 11(7), 2027–2054, doi:10.5194/bg-11-2027-2014, 2014.

Ulzen, J., Abaidoo, R. C., Mensah, N. E., Masso, C. and AbdelGadir, A. A. H.: Bradyrhizobium inoculants enhance grain yields of soybean and cowpea in Northern Ghana, Front. Plant Sci., 7, 1–9, doi:10.3389/fpls.2016.01770, 2016.

Vanlauwe, B., Hungria, M., Kanampiu, F. and Giller, K. E.: The role of legumes in the sustainable intensification of African smallholder agriculture: Lessons learnt and challenges for the future, Agric. Ecosyst. Environ., 284(December), 106583, doi:10.1016/j.agee.2019.106583, 2019.

Waha, K., Van Bussel, L. G. J., Müller, C. and Bondeau, A.: Climate-driven simulation of global crop sowing dates, Glob. Ecol. Biogeogr., 21(2), 247–259, doi:10.1111/j.1466-8238.2011.00678.x, 2012.

Waring, R. H., Landsberg, J. J. and Williams, M.: Net primary production of forests: A constant fraction of gross primary production?, Tree Physiol., 18(2), 129–134, doi:10.1093/treephys/18.2.129, 1998.

Wu, L. and McGechan, M. B.: Simulation of nitrogen uptake, fixation and leaching in a grass/white clover mixture, Grass Forage Sci., 54(1), 30–41, doi:10.1046/j.1365-2494.1999.00145.x, 1999.

Zapata, F., Danso, S. K. A., Hardarson, G. and Fried, M.: Time Course of Nitrogen Fixation in Field-Grown Soybean Using Nitrogen-15 Methodology, Agron. J., 79(1), 172–176, doi:10.2134/agronj1987.00021962007900010035x, 1987.

Zhang, B., Tian, H., Lu, C., Dangal, S., Yang, J. and Pan, S.: Manure nitrogen production and application in cropland and rangeland during 1860–2014: A 5-minute gridded global data set for Earth system modeling, Earth Syst. Sci. Data, 9, 667–678, doi:10.5194/essd-2017-11, 2017.